# Pathfinding quantum simulations of neutrinoless double-$\beta$ decay

Ivan A. Chernyshev [1], Roland C. Farrell[2], Marc Illa [3], Martin J. Savage [3] ✉, Andrii Maksymov[4], Felix Tripier[4], Miguel Angel Lopez-Ruiz[4], Andrew Arrasmith[4], Yvette de Sereville[4], Aharon Brodutch[4], Claudio Girotto[4], Ananth Kaushik[4] & Martin Roetteler[4] ✉

We present results from co-designed quantum simulations of the neutrinoless double-$\beta$ decay of a simple nucleus in 1+1D quantum chromodynamics using IonQ's Forte-generation trapped-ion quantum computers. Electrons, neutrinos, and up and down quarks are distributed across two lattice sites and mapped to 32 qubits, with an additional 4 qubits used for flag-based error mitigation. A four-fermion interaction is used to implement weak interactions, and lepton-number violation is induced by a neutrino Majorana mass. Quantum circuits that prepare the initial nucleus and time evolve with the Hamiltonian containing the strong and weak interactions are executed on IonQ Forte Enterprise. Enabled by tuned model parameters, lepton-number violation is observed in real time, providing a clear signal of neutrinoless double-$\beta$ decay. This was made possible by co-designing the simulation to maximally utilize the all-to-all connectivity and native gate-set available on IonQ's quantum computers. Quantum circuit compilation techniques and co-designed error-mitigation methods, informed from executing benchmarking circuits with up to 2,356 two-qubit gates, enabled observables to be extracted with high precision. We discuss the potential of future quantum simulations to provide yocto-second resolution of the reaction pathways in these, and other, nuclear processes.

Future quantum computers are expected to enable ab initio investigations of key unsolved problems in nuclear physics (NP) and high-energy physics (HEP) research[1–10]. A central focus in the current era of noisy intermediate-scale quantum (NISQ)[11] computers is the co-design of efficient quantum algorithms, circuit compilers and error mitigation schemes to progress toward addressing specific scientific objectives. These advances aim to maximize the utility of quantum computers that feature imperfect gates acting on a limited number of qubits (or qudits). Such NISQ devices are still powerful tools for simulating out-of-equilibrium dynamics, e.g., refs. 12–33, and can provide valuable insight into mechanisms underlying quantum many-body phenomena.

An intriguing future application of quantum computers is the simulation of nuclear reactions in real-time, providing temporal snapshots of nuclei during decays, fission, fusion, and more. These processes involve characteristic time-scales separated by many orders of magnitude. Such multi-scale problems present significant challenges, even for quantum computers. In this work, we propose using quantum computers to image nuclear dynamics on the shortest of these time-scales; a yocto-second ($10^{-24}$ seconds $\equiv$ 1ys). We do not consider high-energy probes, such as beyond-TeV scale collisions, in this discussion. This is the time scale relevant to hadronic structure with, e.g., the $\Delta$-resonance decaying to a proton with a half-life $\tau_\Delta \sim$ 5ys.

[1]Theoretical Division, Los Alamos National Laboratory, Los Alamos, NM, USA. [2]Institute for Quantum Information and Matter (IQIM) and Department of Physics, California Institute of Technology, Pasadena, CA, USA. [3]InQubator for Quantum Simulation (IQuS), Department of Physics, University of Washington, Seattle, WA, USA. [4]IonQ Inc., College Park, MD, USA. ✉e-mail: mjs5@uw.edu; martin.roetteler@ionq.co

The analogous (experimental) development of femto-second ($10^{-15}$ s) imaging in the 1990s[34] gave chemists access to the intermediate states that molecules pass through during chemical reactions, and revealed how atoms re-arrange during the breaking and formation of chemical bonds. The probing of sub-yocto-second dynamics using quantum simulations would provide analogous insight(s) into nuclear processes. Snapshots of the quantum state of nuclei on these extremely short time scales are, in principle, accessible via Hamiltonian evolution on a quantum computer.

In this work, we investigate a potential exotic nuclear decay relevant to searches for new physics. The stability of matter places tight constraints on the structure of physics beyond the Standard Model[35–40], including upper bounds on the amount that fundamental symmetries can be broken obtained from proton-decay, neutron-antineutron-oscillations, and the $\beta$-decay and $\beta\beta$-decay of nuclei, see, e.g., refs. 41–46. In the case at hand, the neutrinoless double $\beta$-decay ($0\nu\beta\beta$-decay) of certain nuclei can only occur if one of the (accidental) symmetries of the Standard Model (lepton number) is broken. Determining the rates of such decays is a forefront theoretical and computational challenge due to the Majorana-neutrino[47] induced process, requiring two charged-current weak interactions connected by a near-massless neutrino propagating across the nucleus (see, e.g., refs. 48,49). Progress toward robust computations of decay rates of nuclei continues to be impressive[50–56], including with Euclidean-space lattice quantum chromodynamics (QCD) simulations, e.g., refs. 57–65, but keeping track of the coherent sum over low-energy excitations of the nucleus during this process, along with the strong correlations between nucleons, is a task that may be better suited for quantum, rather than classical, computation.

Future simulations aided by quantum computers could help resolve two puzzles in the Standard Model: the nature of the neutrino mass and the mechanism behind the matter/anti-matter asymmetry in our universe. This is because a lepton number violating neutrino mass would be intimately tied to the matter/anti-matter imbalance created during the electroweak phase transition in the early universe. These fundamental questions about nature have motivated an internationally-coordinated experimental program[66–74] searching for the $0\nu\beta\beta$-decay of nuclei, and synergistic theoretical and computational efforts[50,75–77]. Importantly, an experimental observation of $0\nu\beta\beta$-decay would provide unambiguous evidence for the violation of lepton number. A more detailed discussion on the potential scientific insight gained by, and challenges associated with, simulating $0\nu\beta\beta$ is given in Supplementary Note 1.

The purpose of the present work is to establish the current capabilities of trapped-ion quantum computers to simulate a rare process of current experimental and theoretical focus that has the potential to reveal new insights into fundamental physics. Our work is a step along a path that is expected to lead to quantum simulations that can be used to determine or constrain new physics from corresponding experimental results. We leverage the power of co-designed simulations to observe lepton-number violating dynamics on a quantum computer for the first time. This is accomplished via the simulation of the $0\nu\beta\beta$-decay of a simple nucleus in 1+1D lattice QCD using IonQ's Forte-generation quantum computers. Specifically, we perform lattice simulations of the decay of two baryons restricted to two spatial sites. Both strong and weak interactions are included, and a Majorana neutrino mass term explicitly violates lepton-number conservation. The coupling constants and masses are deliberately tuned to recover a mass hierarchy that kinematically favors double-$\beta$ decay, but suppresses single-$\beta$ decay (in this volume).

Our work builds off previous quantum simulations of non-abelian gauge theories in 1+1D dimension[15,78–82] and beyond[17,18,83–90], with a particular emphasis on ref. 79, where single $\beta$-decay was simulated using a similar setup. Our quantum circuits are designed to maximally benefit from the all-to-all connectivity and native gate-set available on IonQ's trapped-ion quantum computers. Additionally, we introduce techniques for mitigating statistical and device errors that are tailored to be maximally effective for the observables we measure. These co-designed elements enable high-fidelity real-time simulations of doubly-weak decays on a two spatial site (32 qubit) lattice. A $10\sigma$ signal for the dynamical generation of lepton-number violation mediated by a Majorana neutrino is obtained from running circuits with 470 two-qubit gates on IonQ Forte Enterprise. This work establishes a potential path forward for future quantum simulations that would impact searches for new physics.

## Results

### $0\nu\beta\beta$-decay in 1+1D QCD

A model for the $0\nu\beta\beta$-decay of a nucleus is simulated in 1+1D lattice QCD with dynamical quarks (up and down) and leptons (electrons and neutrinos). A lattice with periodic boundary conditions (PBCs) and $L = 2$ spatial lattice sites is mapped to 32 qubits of IonQ's Forte-generation quantum computers. A minimum of two spatial sites is needed to support the degrees of freedom produced in $0\nu\beta\beta$-decay. The weak interactions are modeled with an effective four-Fermi interaction that locally couples quarks to leptons[79]. The hadronic states of 1+1D QCD form isospin multiplets[78], with the lowest-lying baryon multiplet having $I = 3/2$, containing $\Delta^{++}, \Delta^+, \Delta^0, \Delta^-$ labeled after its similarity with the $\Delta$ resonance in 3+1D QCD (the superscripts denote the electric charge). Parameters in the Hamiltonian, including a Majorana mass term, are tuned to permit the $0\nu\beta\beta$-decay of a $|\Delta^-\Delta^-\rangle$ initial state.

To simulate this decay, a quantum circuit that initializes the lepton vacuum and $|\Delta^-\Delta^-\rangle$ is applied, and then time evolved for time t with two Trotter steps of a Hamiltonian containing the strong and weak interactions, as well as the free fermion terms. The inclusion of a lepton-number breaking neutrino Majorana mass term in the Hamiltonian $\hat{H}_{\text{Maj}}$ opens the $0\nu\beta\beta$ decay channel.

To reduce the number of two-qubit gates, we implement several approximations in the time evolution. The chromoelectric interaction is truncated beyond $\lambda = 1$ staggered sites and we only keep the terms in the four-Fermi weak interaction, $\hat{H}_\beta^{1+1}$, that act on valence fermions. Additionally, two-qubit rotations with angles $\theta \leq t/32$ are removed. The effects of these approximations are detailed in Supplementary Note 2. Despite the errors due to approximation becoming significant for $t \geq 1.0$, our simulations are still able to extract qualitatively correct signals of $0\nu\beta\beta$-decay. After using IonQ's circuit compiler, the required circuits have 470 $R_{ZZ}$ gates. A schematic of the quantum circuit(s) used in our simulations is shown in Fig. 1.

Weak decays can be detected and classified by measuring the total electric charge of the electrons $\hat{Q}_e$ and the lepton number $\hat{\mathcal{L}}$. These observables are

$$\hat{Q}_e = -\frac{1}{2}\sum_{n=0}^{3}\hat{Z}_{25+2n},$$

$$\hat{\mathcal{L}} = \frac{1}{2}\sum_{n=0}^{3}\left(\hat{Z}_{24+2n} + \hat{Z}_{25+2n}\right). \tag{1}$$

Both the lepton electric charge and lepton number are zero in the initial state, and a non-zero lepton electric charge signals a decay. Further, deviations of the lepton number from zero are due to the neutrino Majorana mass. While not independent from these two quantities, the neutrino number $\hat{N}_\nu = \hat{Q}_e + \hat{\mathcal{L}}$ can be helpful in revealing the contributions from neutrinoless decays when the Majorana mass is non-zero, $m_M \neq 0$.

### Estimating the limits of IonQ Forte

In the spirit of co-design and benchmarking, we present results from simulations of $0\nu\beta\beta$-decay that pushed against the outer limits of

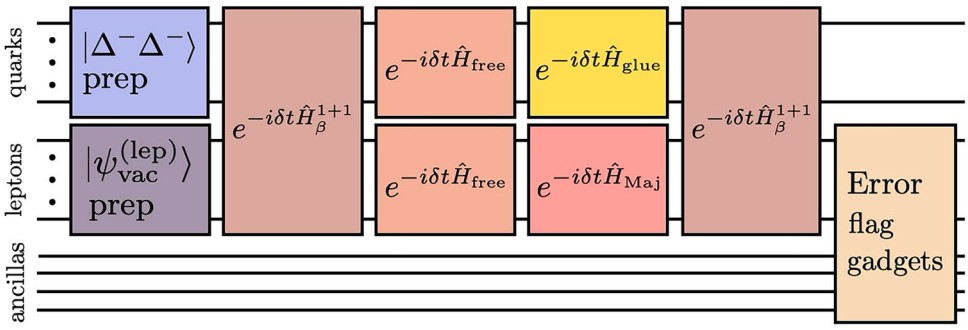

**Fig. 1 | The structure of the quantum circuits used to simulate $0\nu\beta\beta$-decay.** First, the initial $|\Delta^-\Delta^-\rangle$ state is prepared using SC-ADAPT-VQE[118]. Next, weak decay dynamics are implemented using two steps of Trotterized time evolution with time step $\delta t = t/2$. The first Trotter step has been simplified because $|\Delta^-\Delta^-\rangle$ is a strong interaction eigenstate. Before the final measurement, leakage events are flagged by coupling the lepton qubits to a register of ancillas. All qubits are initialized in $|0\rangle$ and measured in the z-basis. Decompositions of each circuit block are provided in Supplementary Notes 10 and 5.

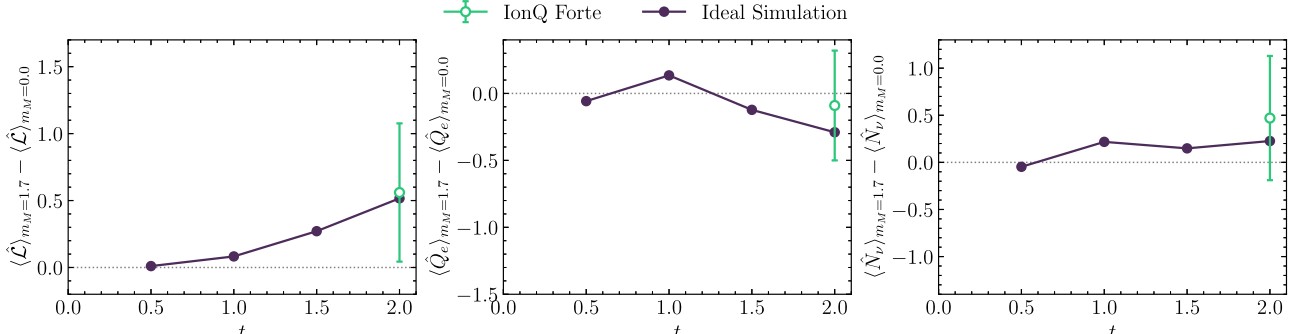

**Fig. 2 | The time evolution of the difference of the lepton number (left), lepton electric charge (center) and neutrino number (right) during the decay of the $|\Delta^-\Delta^-\rangle$ two-baryon state in 1+1D QCD.** Results are obtained from two steps of the first-order Trotterized circuit (2356 native two-qubit gates) executed on IonQ Forte, and are derived from those given in Table 1, displayed in green, as well as noiseless statevector simulator (Ideal Simulation), displayed in black. Both sea- and valence-weak interactions are included. The error bars are obtained from bootstrap resampling and represent one standard deviation. The gray dotted line is added for reference.

**Table 1 | Observables at $t = 2$ in the decay of the $|\Delta^-\Delta^-\rangle$ initial state obtained from a noiseless statevector simulator (Ideal Simulation) and from executing quantum circuits with 2356 native two-qubit gates on IonQ Forte (QPU results)**

| Ideal Simulation | | | | QPU Results | | | |
|---|---|---|---|---|---|---|---|
| $\langle\widehat{\mathcal{L}}\rangle_{m_M=0}$ | $\langle\widehat{\mathcal{L}}\rangle_{m_M=1.7}$ | $\langle\widehat{Q}_e\rangle_{m_M=0}$ | $\langle\widehat{Q}_e\rangle_{m_M=1.7}$ | $\langle\widehat{\mathcal{L}}\rangle_{m_M=0}$ | $\langle\widehat{\mathcal{L}}\rangle_{m_M=1.7}$ | $\langle\widehat{Q}_e\rangle_{m_M=0}$ | $\langle\widehat{Q}_e\rangle_{m_M=1.7}$ |
| 0.0 | 0.57 | −0.67 | −0.76 | − 0.02 ± 0.36 | 0.54 ± 0.37 | − 0.13 ± 0.29 | − 0.22 ± 0.29 |

The uncertainties are estimated from bootstrap resampling and represent one standard deviation. Differences between these results are shown in Fig. 2.

cutting-edge quantum computers. Specifically, we probed the boundaries of IonQ Forte's capabilities by executing circuits with ~ 5 × more two-qubit gates. These experiments simulated $0\nu\beta\beta$-decay with fewer approximations applied to the time-evolution operator. The full four-Fermi weak interaction, including both valence and sea components, given in Supplementary Note 3A, was implemented and small rotation angles were retained. The required quantum circuits had 2,356 two-qubit gates compared to the 470 two-qubit gates executed on Forte Enterprise. A total of 24 twirled variants (see Methods' subsection 'Circuit-Optimization and Error-Mitigation'), each with 420 shots, were run, leading to a total of 10,080 shots.

Results obtained from Forte are shown in Fig. 2 for $t = 2.0$, and given in Table 1. Compared to the experiments run on Forte Enterprise in Results IIC, the uncertainties from Forte are significantly larger due to the increased noise in deeper circuits. The uncertainties estimated from bootstrap resampling are known to be (very) conservative as they include elements of the hardware-bias cancellation (while the mean values are close, the uncertainty from bootstrapping is much larger

still than that from the un-bootstrapped result). This leads to a deviation of the bootstrapped mean from the true mean, and illustrates that, in this instance, bootstrap resampling provides biased estimators.

From these simulations, we concluded that implementing the full weak operator was not practical with the current generation of hardware. The quantum circuits were too deep for error mitigation to applied successfully. Consequently, we reduced the circuit depth by truncating the weak operator to only include valence fermion operators for subsequent simulations. These runs also informed the error mitigation strategy utilized on IonQ Forte Enterprise as discussed in IV D.

### Observation of $0\nu\beta\beta$-decay in 1+1D QCD using IonQ Forte Enterprise

Results obtained from quantum simulations using IonQ Forte Enterprise are shown in Fig. 3 for $t = \{0.5, 1.0, 1.5, 2.0\}$ and two values of the Majorana mass $m_M = \{0.0, 1.7\}$, as well as in Table 2. For times

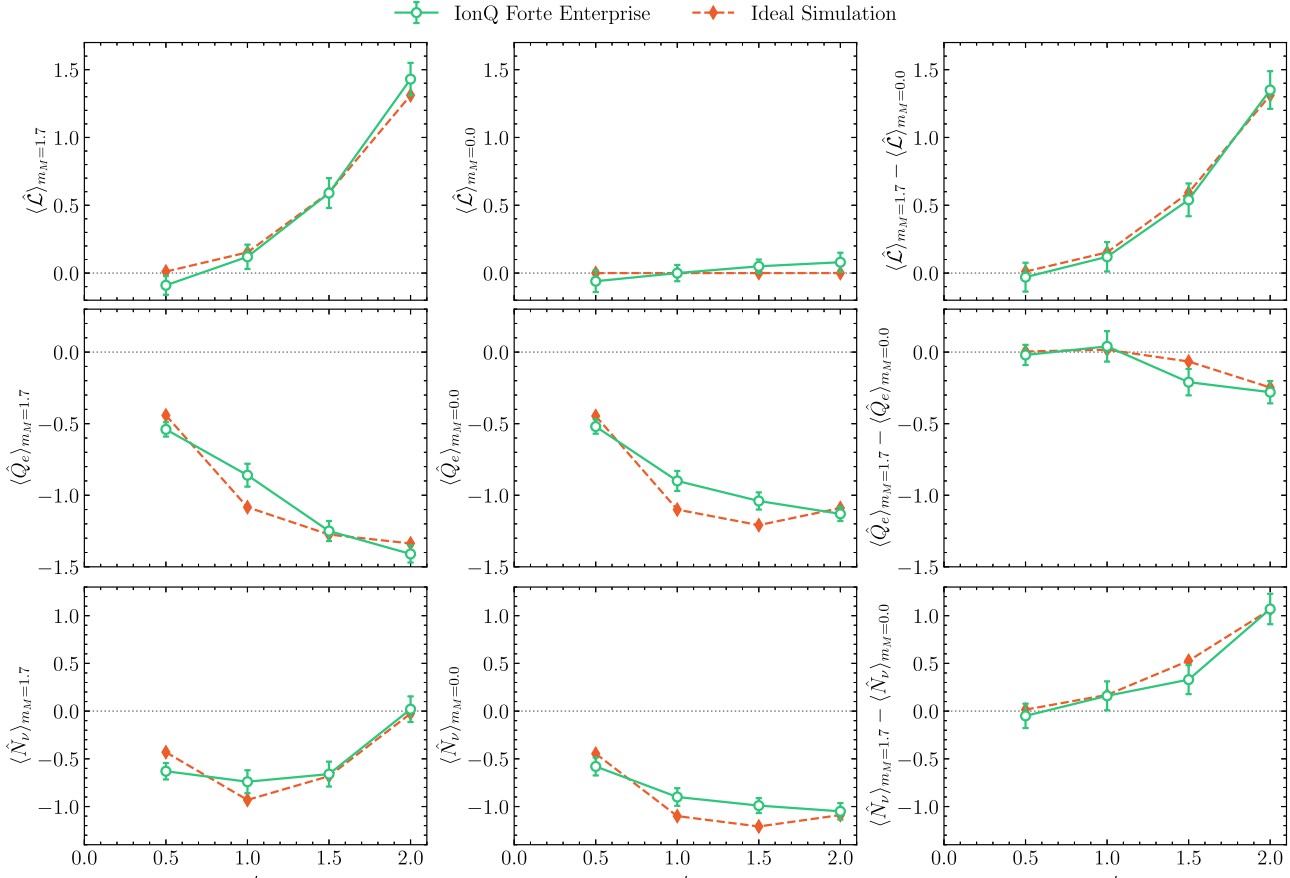

**Fig. 3 | The time evolution of the lepton number (upper row), lepton electric charge (middle row), and neutrino number (lower row) during the decay of the** $|\Delta^-\Delta^-\rangle$ **two-baryon state in 1+1D QCD.** Two steps of first-order Trotterized time evolution using the (approximate) valence-fermion weak interactions are implemented, requiring 470 two-qubit gates. The left panels show the results obtained with a Majorana mass of $m_M = 1.7$, the center panels show results for $m_M = 0$, and the

right panels show the differences between the $m_M = 1.7$ and $m_M = 0$ results. The green points were obtained from IonQ Forte Enterprise, and the orange diamonds correspond to noiseless simulation. The error bars are obtained from bootstrap resampling and represent one standard deviation, and the gray dotted line is added for reference.

**Table 2 | The lepton number and electric charge obtained from a noiseless statevector simulator (Ideal Simulation) and from Forte Enterprise (QPU results)**

| | Ideal Simulation | | | | QPU Results | | | |
|---|---|---|---|---|---|---|---|---|
| $t$ | $\langle \widehat{\mathcal{L}} \rangle_{m_M = 0}$ | $\langle \widehat{\mathcal{L}} \rangle_{m_M = 1.7}$ | $\langle \widehat{Q}_e \rangle_{m_M = 0}$ | $\langle \widehat{Q}_e \rangle_{m_M = 1.7}$ | $\langle \widehat{\mathcal{L}} \rangle_{m_M = 0}$ | $\langle \widehat{\mathcal{L}} \rangle_{m_M = 1.7}$ | $\langle \widehat{Q}_e \rangle_{m_M = 0}$ | $\langle \widehat{Q}_e \rangle_{m_M = 1.7}$ |
| 0.5 | 0.0 | 0.01 | − 0.45 | − 0.44 | − 0.06 ± 0.08 | − 0.09 ± 0.07 | − 0.52 ± 0.05 | − 0.54 ± 0.05 |
| 1.0 | 0.0 | 0.15 | − 1.10 | − 1.08 | 0.00 ± 0.06 | 0.12 ± 0.09 | − 0.90 ± 0.07 | − 0.86 ± 0.08 |
| 1.5 | 0.0 | 0.59 | − 1.20 | − 1.26 | 0.05 ± 0.05 | 0.59 ± 0.11 | − 1.04 ± 0.06 | − 1.25 ± 0.07 |
| 2.0 | 0.0 | 1.31 | − 1.09 | − 1.34 | 0.08 ± 0.07 | 1.43 ± 0.12 | − 1.13 ± 0.05 | − 1.41 ± 0.06 |

The QPU results correspond to the green points (mean value and ± 1σ uncertainty) in Fig. 3. Non-linear filtering has been applied to the raw results, which are also post-selected on the conservation of total electric charge, red, green, and blue color charges, as well as no leakage detected by the flag qubits.

$t$ = {0.5, 1.0, 1.5} these results were derived from 14,400 shots, while 24,000 were taken for time $t$ = 2.0. Approximately 10% of the shots survived after filtering on leakage detection and conservation of color and total electric charges. The error bars on these results were computed by bootstrap resampling, see Supplementary Note 4 for details.

A clear (statistically significant) signal for the violation of lepton number is found for $m_M = 1.7$, as shown in Fig. 3, which is absent for $m_M = 0$. Specifically, at $t$ = 2.0, there is a 10σ difference between the lepton numbers obtained with $m_M = 1.7$ and $m_M = 0$. The ideal simulation results obtained from noiseless simulation using classical computers are also shown in Fig. 3, and are in excellent agreement with the results obtained from Forte Enterprise. The effects of the different

error mitigation methods applied during the simulations are shown in Supplementary Table 5 in Supplementary Note 5. The impact of the combined error-mitigation on the raw results obtained from the device are essential in obtaining reliable results (as determined by comparison with the classically computed counterparts). The flag gadgets do not make a statistically significant difference to the results, changing central values by less than 1σ at each time step. On the other hand, post-selection is essential.

The lepton electric charge and lepton number in Fig. 3 (and extended to later times in Supplementary Note 2) do not exhibit the expected time dependence of an exponential decay. This is a finite-size effect, and an exponential decay is expected to emerge in the

continuum and infinite volume limits, where the density of states becomes sufficiently dense at the kinematics of the transition energy. This was studied in detail in Appendix D of ref. 79.

For $m_M = 0$, while the lepton number remains consistent with zero at all times, as expected, electric charge is produced in the lepton sector. This is consistent with single $\beta$-decay, $2\nu\beta\beta$-decay, and transitions to other intermediate states that are energetically disfavored. The baryons and leptons cannot separate after a decay due to the small simulation volume, and the continual interactions between baryons and leptons causes there to be a non-zero lepton electric charge density at all times.

The time dependence of lepton number and lepton electric charge, as displayed in Fig. 3, could, in principle, reveal aspects of the underlying mechanism of the decay process. Initially, the lepton sector has the quantum numbers of the vacuum, $\mathcal{L} = Q_e = 0$. The expectation value of these quantities change in time due to the decay of the $|\Delta^- \Delta^-\rangle$ di-baryon. For simplicity, consider the two possible time-orderings of two single $\beta$-decays and one Majorana mass term $\hat{H}_{\text{Maj}}$, the minimal operator structure required for $0\nu\beta\beta$-decay to occur. There will be contributions from an arbitrary number of insertions of these operators during unitary time evolution. However, in nature, contributions from multiple insertions are suppressed by the smallness of the weak coupling constant and Majorana mass. In terms of the time-ordered sequence of interaction terms:

- $\hat{H}_\beta^{1+1}$, then $\hat{H}_{\text{Maj}}$, then $\hat{H}_\beta^{1+1}$: $\hat{Q}_e = 0$, $\mathcal{L} = 0$, then $\hat{Q}_e = -1$, $\mathcal{L} = 0$, then $\hat{Q}_e = -1$, $\mathcal{L} = 2$, then $\hat{Q}_e = -2$, $\mathcal{L} = 2$.
- $\hat{H}_\beta^{1+1}$, then $\hat{H}_\beta^{1+1}$, then $\hat{H}_{\text{Maj}}$: $\hat{Q}_e = 0$, $\mathcal{L} = 0$, then $\hat{Q}_e = -1$, $\mathcal{L} = 0$, then $\hat{Q}_e = -2$, $\mathcal{L} = 0$, then $\hat{Q}_e = -2$, $\mathcal{L} = 2$.

Note that the sequence with $\hat{H}_{\text{Maj}}$ acting first does not contribute because the initial state is an eigenstate of the Hamiltonian without $\hat{H}_\beta^{1+1}$. These, and higher order, reaction pathways interfere during the decay process and need to be coherently summed together to predict $\mathcal{L}$ and $\hat{Q}_e$. This coherent evolution of all possible reaction pathways is one of the principle advantages of using quantum computers. Given strong interaction times scales, it is clear that wavefunction evolution at the yocto-scale has the potential to provide key information that can be used to identify dominant decay pathways.

## Discussion

We have performed a suite of path-finding quantum simulations of $0\nu\beta\beta$-decay in 1+1D QCD induced by a lepton-number violating Majorana neutrino mass. The simulations were performed on two lattice sites with two-flavors of quarks and leptons, which maps to 32 qubits, and with unphysical values of the Hamiltonian parameters. Multiple facets of our simulations, from circuit design to error mitigation, were co-designed to maximize the performance from IonQ's Forte-generation trapped-ion quantum computers. This included using extra qubits as ancillae to detect leakage on the most important qubits, and optimizing circuits for Forte's all-to-all connectivity and native $R_{ZZ}$ two-qubit gates.

A production run on IonQ Forte implementing the full weak operator requiring 2,356 two-qubit gates established the limits of the initial co-design and helped to identify effective error-mitigation strategies. Informed by the results of these computations, circuits with 470 two-qubit gates were executed on Forte Enterprise achieving sufficient fidelity to establish a $10\sigma$ signal for the dynamical generation of lepton number violation in $0\nu\beta\beta$-decay induced by the valence weak operator. This demonstrated the neutrinoless decay mechanism of a two-baryon initial state that was only possible due to a non-zero neutrino Majorana mass.

Our work establishes a set of benchmarks for further quantum simulations of exotic weak decays that may eventually impact experimental searches for new physics. Simulating doubly-weak decay processes is a significant challenge for classical computing due to the necessity of coherently tracking the dynamics in a strongly-interacting nucleus. Quantum simulations are expected to eventually improve upon results obtained with classical computation, and provide insight into the underlying strong-interaction mechanisms and pathway(s) at yocto-second and longer time scales.

The next generation of quantum simulations will benefit from improvements in both fidelity and qubit count expected in the next few years[91]. Notably, by the end of 2027, quantum error correction is anticipated to be implemented on IonQ's devices with $\approx 10{,}000$ physical qubits and $\approx 800$ logical qubits. Today's quantum error correction on trapped ion qubits exhibits gate infidelities of as good as $\approx 5 \times 10^{-6}$ for Clifford gates[92] and gate infidelities as good as $\approx 5 \times 10^{-4}$[93] for non-Clifford gates, but IonQ's roadmap aims for logical gate infidelities as good as $10^{-7}$ or less. These advancements will enable progress toward more realistic simulations of exotic weak decays that are required to begin making contact with experiment, complementing an analogous path for lattice QCD combined with effective field theory that has been outlined in refs. 57–65.

Next steps along this path are to extend these simulations to 2+1D, building upon recent progress on efficient Hamiltonian formulations of lattice gauge theories[89,94–96], to increase the size of the spatial volumes using larger registers of qubits, to work closer to the continuum limit, to work with fermion masses closer to their experimental values, and to perform simulations using a range of neutrino masses. In the nearer-term, first steps toward the aforementioned achievements should be feasible with the release of the next-generation IonQ device, Tempo, and the repetition of this study with the full, unapproximated 1+1D QCD and decay Hamiltonian should be feasible shortly afterward. Access to larger 1+1D lattices will allow the leptons emitted during $0\nu\beta\beta$-decay to fully separate from the nucleus, which is important for mitigating finite size effects.

As these quantum simulations become more sophisticated, it will be important to robustly quantify lattice spacing artifacts and extrapolate to infinite volume and physical parameters. Our results illuminate some of the challenges that lie ahead in approaching the physical hierarchy of mass scales that contribute to this type of decay process. The physical strong-interaction energy scales are in the GeV region, while the neutrino masses are sub-eV. Naive methods of simulation will not be possible, even with quantum computers, simply because of the huge discrepancy in the relative time-scales that are involved. Similarly, another challenge will be the robust preparation of the initial-state nucleus in a large spatial volume. This is a difficult problem, even in classical lattice QCD simulations[97–99], due to the excitation energies of nuclei being orders of magnitude smaller than their mass.

Despite these challenges, there are still several advantages expected from quantum simulations. One is to provide insight into the underlying strong-interaction mechanisms and pathway(s) at yocto-second and longer time scales. Once identified, these insights could motivate new, highly efficient, approximation schemes that would improve traditional calculations using classical computers. Additionally, quantum simulations can be performed with a selection of parameters away from the physical point, where the hierarchy between the neutrino mass and strong-interaction scale is not so severe. With a sufficiently large ensemble of simulations, robust extrapolations to the physical parameters will be possible.

## Methods
### The simulation setup
In our simulations, fermions are placed on the lattice using a staggered discretization that maps $L$ spatial sites to $N = 2L$ staggered sites and $2LN_s$ fermion sites, where $N_s$ is the number of fermion species (for more details, see Supplementary Note 3). The Hamiltonian that is used has four terms,

$$\hat{H} = \hat{H}_{\text{free}} + \hat{H}_{\text{glue}} + \hat{H}_{\beta,\text{valence}}^{1+1} + \hat{H}_{\text{Maj}}. \tag{2}$$

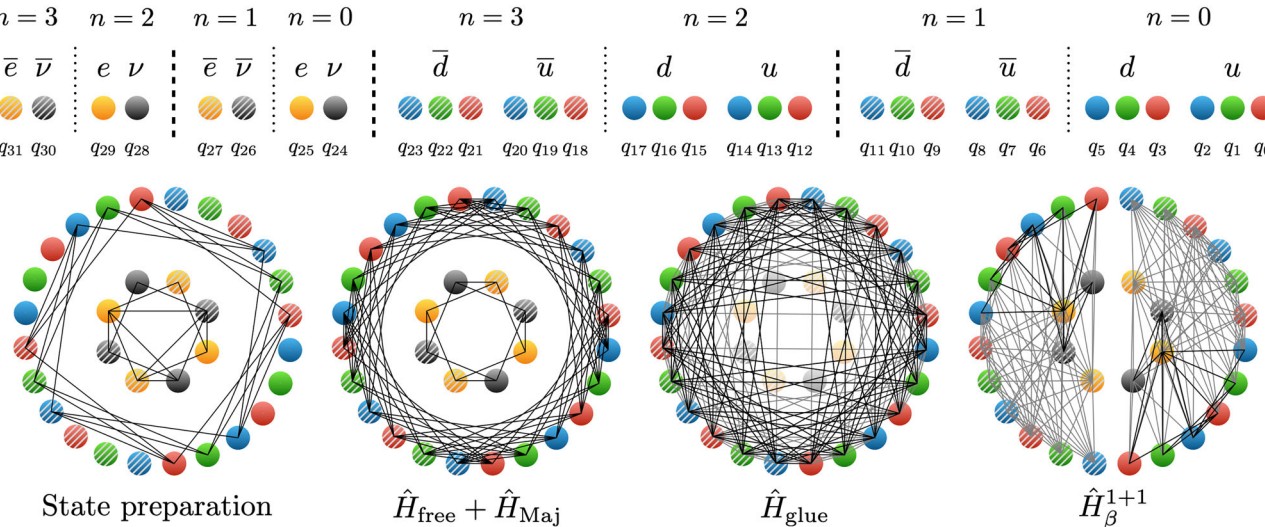

**Fig. 4 | On top, the *L* = 2 lattice-to-qubit mapping used in the 0*νββ*-decay simulations on IonQ's quantum computers.** Spatial sites are separated by dashed lines, while staggered fermion sites (*n* = 0, 1, 2, 3) are separated by dotted lines. Qubits are indexed from right-to-left, with the rightmost qubit being qubit 0, $q_0$. Qubits $q_0, q_1, \ldots, q_{23}$ correspond to up (*u*) and down (*d*) quarks, each of three colors *r, g, b*. Qubits $q_{24}, q_{25}, \ldots, q_{31}$ correspond to the electrons (*e*) and neutrinos (*ν*). The ordering of the lepton and quark spatial sites has been chosen to reduce the length of Pauli strings in the Jordan-Wigner mapping. The bottom shows the required connectivity for state preparation and the Trotterized time evolution for each part of the Hamiltonian. The gray connections are removed after the approximations described in the main text.

The Hamiltonian describing free staggered fermions is[100,101]

$$\widehat{H}_{\text{free}} = \sum_f \sum_{n=0}^{2L-1} \Big[ m_f(-1)^n \phi_n^{(f)\dagger} \phi_n^{(f)} + \frac{1}{2}\Big(\phi_n^{(f)\dagger} \phi_{n+1}^{(f)} + \text{h.c.}\Big) \Big], \tag{3}$$

where $m_f$ are the bare (Dirac) fermion masses with $f \in \{u_r, u_g, u_b, d_r, d_g, d_b, e, \nu\}$ corresponding to red, green and blue up and down quarks, and the two leptons, electrons and neutrinos. The index *n* labels the staggered site, with *n* even corresponding to fermion sites (quarks and leptons), and *n* odd to anti-fermion sites (anti-quarks and anti-leptons).

The QCD interactions are encoded in $\widehat{H}_{\text{glue}}$. Explicit gauge degrees of freedom are absent in axial gauge[78,81], leaving a non-linear color Coulomb interaction between the quarks. With PBCs, the interaction is[102]

$$\widehat{H}_{\text{glue}} = \frac{g^2}{2} \sum_{n=0}^{2L-1} \sum_{s=1}^{\lambda} \Big(-s + \frac{s^2}{2L}\Big)\Big(1 - \frac{1}{2}\delta_{s,L}\Big) \sum_{a=1}^{8} Q_n^{(a)} Q_{n+s}^{(a)}, \tag{4}$$

where *g* is the QCD interaction strength and

$$Q_n^{(a)} = \sum_{f=u,d} \phi_n^{(f)\dagger} T^{(a)} \phi_n^{(f)}, \tag{5}$$

are the eight *SU*(3) charges (color indices have been suppressed). The dynamics of the zero-mode of the gauge field is not considered in this work, see Supplementary Note 3. Recent work has shown that the mechanism of confinement motivates an approximate interaction of these naively long-range interactions[22]. This approximation truncates interactions beyond *λ* staggered sites, and converges exponentially for *λ* larger than the confinement scale. For the simulation parameters selected in this work, *λ* = 1 is found to be sufficient, and will be used throughout.

Weak interactions are modeled through a local vector-like four-Fermi operator[79],

$$\widehat{H}_{\beta,\text{valence}}^{1+1} = \frac{G}{\sqrt{2}} \sum_{n \text{ even}} \Big(\phi_n^{(u)\dagger} \phi_n^{(d)} \phi_n^{(e)\dagger} \phi_{n+1}^{(\nu)} + \text{h.c.}\Big), \tag{6}$$

where *G* is the weak coupling constant (Fermi's constant). This is an early-time approximation of the full four-Fermi operator that retains only the terms acting on the "valence" quarks and leptons[79]. Similar operator truncations were used in early quenched lattice QCD calculations, e.g., ref. 103. The impact of retaining terms beyond the valence sector are considered in Supplementary Note 3C. Note that color indices, which are summed over, have been suppressed. This approximation will be used to reduce the number of two-qubit gates required for time evolution (see Supplementary Note 2C).

Lastly, the neutrino Majorana mass term is[79]

$$\widehat{H}_{\text{Maj}} = \frac{1}{2} m_M \sum_{n \text{ even}} \Big(\phi_n^{(\nu)} \phi_{n+1}^{(\nu)} + \text{h.c.}\Big). \tag{7}$$

This is the unique local fermionic operator that violates lepton number by two units while preserving the other symmetries.

The Jordan-Wigner transformation (JW)[104] is used to map the fermionic Hilbert space to 16*L* qubits. The simulations in this work will be performed on *L* = 2 spatial sites, with the qubit-to-lattice layout shown in Fig. 4. All of the terms in $\widehat{H}$ are local on the level of spatial sites, but can become long-range operations when mapped to qubits. The non-trivial connectivity required to perform all the steps in the simulation is shown in the lower panels of Fig. 4. Because of this, the native all-to-all connectivity hosted by IonQ trapped-ion quantum computers is essential for efficient simulations. The Hamiltonian expressed in terms of spin operators is given in Supplementary Note 3.

Hadronic weak decays begin with hadrons in the QCD sector and the vacuum in the lepton sector. We choose to initialize a (maximally

isospin stretched) two-baryon state, "the $\Delta^-\Delta^-$ dibaryon",

$$\left|\psi_{\text{init}}\right\rangle = \left|\psi_{\text{vac}}^{(\text{lep})}\right\rangle \left|\Delta^-\Delta^-\right\rangle . \tag{8}$$

For $L = 2$, the quark wavefunction factorizes into a tensor product of the one-flavor up-quark vacuum and a fully occupied $d$-quark register[79]. This factorization simplifies the quantum circuits needed for state preparation.

Current experimental searches for $0\nu\beta\beta$-decay involve atoms where $\beta$-decay is kinematically forbidden, but $\beta\beta$-decay is allowed. This hierarchy occurs naturally in, for example, $^{76}$Ge, and produces a clear signal of $\beta\beta$-decay. In our model, we reproduce this by tuning the masses and coupling constants to engineer the desired hierarchy in the spectrum. Specifically, our simulations use

$$m_u = 1 , \; m_d = 1.5 ,$$
$$m_e = 0.1 , \; m_\nu = 1.5 , \; m_M = \{0.0, 1.7\} , \tag{9}$$
$$g = 1 , \; G = 1 .$$

This choice of parameters makes $\beta$-decay energetically disfavored, while the lepton number violating decay

$$\left|\Delta^-\Delta^-\right\rangle \; \rightarrow \; \left|\Delta^0\Delta^0\right\rangle + 2e^- \tag{10}$$

is not hindered by kinematical constraints. This is the process we identify in our quantum simulations. Additional details on the spectrum are given in Supplementary Note 6. We emphasize that the parameters defining the 1+1D Hamiltonian implemented in this work are not related to those in nature. They are selected to enable a model simulation of $\beta\beta$-decay that is a first step toward future simulations in this genre of fundamental physics. Supplementary Note 7 provides a short discussion of future extrapolations that will be required to be able to make predictions for physical observables.

### Overview of IonQ's trapped-ion quantum computers

In this work we made use of two of IonQ's Forte-generation quantum processing units (QPUs)[105], Forte and Forte Enterprise. These systems are very similar, with Forte Enterprise being the second device to be manufactured and made available with the Forte architecture. In these systems, 36 qubits are realized as trapped $^{171}$Yb$^+$ ions, with quantum information encoded in two hyperfine levels of the ground state. Ions are sourced via laser ablation and selective ionization before being loaded into a surface linear Paul trap in a compact integrated vacuum package. Manipulation of the qubit states is achieved by illuminating individual ions with pulses of 355 nm light that drive two-photon Raman transitions, thereby enabling the implementation of arbitrary single-qubit rotations and two-qubit $R_{ZZ}$ entangling gates. The median direct randomized benchmarking (DRB)[106] fidelities of the entangling gates at the time of execution were 99.3% on Forte and 99.5% on Forte Enterprise, with gate durations around 950 μs.

IonQ Forte-generation devices integrate acousto-optic deflectors (AOD) that allow for independent steering of each laser beam to its respective ion, substantially reducing beam alignment errors across the ion chain[107,108]. This optical architecture, combined with a robust control system that automates calibration and optimizes gate execution, has enabled the realization of larger qubit registers with enhanced gate fidelities.

### Circuit-optimization and error-mitigation

When working with a quantum computer with limited resources, it is important to design the quantum circuits and error handling in tandem. On NISQ devices, this amounts to careful circuit optimization and error mitigation techniques[109]. Quantum error-mitigation techniques make use of methods like symmetrization, twirling[110], amplification and extrapolation, regression, and/or post-selection in order to arrive

at an approximation of the ideal circuit outputs for a given application. Noise-induced biases can be removed by leveraging application and device symmetries, with a smaller overhead cost than with random sampling. This approach can be especially efficient when the device-level biases are known[111]. Combining the careful selection of circuit-variant implementations with observable-specific post-selection rules allows for precise error detection and higher shot efficiency. The approach chosen here combines debiasing through symmetrization[111], post-selection on symmetry checks, and a novel parametrized non-linear filtering method on the lepton qubit register. Post-selection is based on the usage of spare qubits for flag-based[112] mid-circuit symmetry checks and leakage error detection to further reduce the errors.

The all-to-all connectivity and native $R_{ZZ}(\theta)$ gates available on IonQ's quantum computers is used to further optimize quantum circuits by merging blocks of two-qubit gates. This reduces the number of entangling-gates by 15%. All-to-all connectivity also allows for an efficient reduction in the bias between the different circuit implementations by varying the qubit-to-ion assignment without any additional gates. In addition to qubit remapping, we make use of a type of phase-flip twirling of two qubit gates in generating our variants as described in ref. 111. For this project we also introduced a bit-flip symmetrization of readout into this process. This symmetrization involves generating pairs of circuit variants, where one of the pair applies bit-flips before and after measurement on a random set of qubits. The other variant of the pair is identical except for having the bit-flips before and after measurement applied to the complement of the set of qubits chosen for the first variant.

In post-processing, we combine the post-selected measurement statistics from different twirled variants, filtering out outlier bit strings. This filtering is accomplished by checking if a given measured bit string appears in at least some specified number of variants, referred to as the filter threshold. The choice of the threshold is determined by a combination of knowledge of the device noise, the number of twirled variants, and the number of shots taken per variant. A higher threshold better mitigates hardware-noise induced biases, but requires more variants and shots per variant. Indeed, for a fixed number of variants and shots, a threshold that is set too high will have simple finite sampling effects that can lead to the loss of all information.

To further optimize circuit performance, we used multiple noise tailoring, error mitigation and error detection methods. Details of each method are given in Supplementary Notes 5 and 8. We apply Pauli twirling[113], XY4 dynamic decoupling[114,115] and measurement twirling through bit flipping[116,117]. These techniques mitigate coherent two-qubit over- and under-rotations, phase and idling errors, spontaneous emission and measurement biases. Each circuit is compiled to pairs of twirled variants with unique qubit assignments to debias qubit-associated error dependencies. Each pair of variants is identical up to bit flips before and after measurement that symmetrize readout errors. For the circuits run on Forte Enterprise, a total of 96 twirled variants, each with 150 shots, was chosen to balance the ability to mitigate (systematic) hardware errors by increasing the number of variants, and reducing statistical uncertainties by performing more shots per variant. For $t = 2$, we included 64 additional variants, totaling 160 (24,000 shots total). The tradeoff between number of twirled variants and shots per variant was informed by a Monte Carlo analysis over a uniform distribution of possible output bit string distributions, assuming the hardware error rates taken from benchmarking data. See Supplementary Note 9 for details.

Our simulations only utilized 32 of the 36 qubits available on the Forte-generation machines. On Forte Enterprise, the remaining four qubits are used to flag qubit leakage outside of the computational subspace. See Supplementary Note 5 for more information. Measured bit strings are post selected based on the ancillae states indicating the absence of leakage. Additionally, bit strings are post-selected to conserve color and total electric charges. The charge operators are given

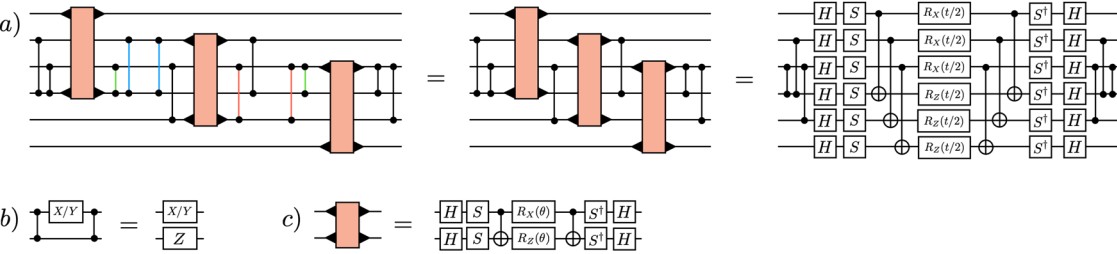

**Fig. 5 | A method for constructing shallow circuits on quantum computers with all-to-all connectivity.** The barbells denote $CZ$ gates, and the ▶-symbols on the orange blocks mark the qubits that are acted on. **a** A quantum circuit that implements the time-evolution of the one-flavor quark kinetic term in Eq. (13). The red, blue, and green $CZ$ gates cancel against each other. **b** A useful circuit identity. **c** The definition of the light orange circuit block that implements $e^{-i\theta(\widehat{\sigma}^+\widehat{\sigma}^- + \widehat{\sigma}^-\widehat{\sigma}^+)}$.

in Supplementary Note 3. On Forte, the 4 extra qubits on the 36 qubit register were used for mid-circuit symmetry iSWAP checks. See Supplementary Note 5 for more details.

### Error mitigation strategies informed from Forte runs

To determine the non-linear filtering threshold, number of variants and the implementation of mid-circuit symmetry checks, smaller test circuits with four-qubit Givens rotations were run. The non-linear filtering threshold was adjusted to have no fewer than 200 surviving counts after filtering. See Supplementary Note 8 for more details. After analyzing the results from the runs on Forte, we noticed that leakage errors make a large contribution. This informed our decision to perform leakage checks instead of iSWAP checks on our second set of runs on Forte Enterprise. For the second set of runs, we kept the same filtering threshold and added post-selection on conserved charges: $r = b = g = 2$ and the total electric charge $Q_{tot} = -2$.

On Forte, we ran reference circuits similar to the ones used in the operator decoherence renormalization (ODR) error-mitigation strategy[22,78,85,87,118,119]. The output of these reference circuits can be efficiently determined using classical computing, and deviations from expected results are used to learn features of the device noise. However, due to the limited number of shots and residual bias in the single-qubit $\widehat{Z}$ observables, we could not apply ODR to the 2,356 two-qubit gate circuits. Instead, we used the reference circuits to inform our noise models. For these experiments, instead of ODR we used debiasing with non-linear filtering, which we found to be more effective with noisier results and smaller numbers of shots. Further, the non-linear filtering works well with post-selection, while ODR is not compatible with it. The information gathered from these limit-testing runs will be valuable in designing the next generation of $0\nu\beta\beta$-decay simulations on trapped-ion platforms.

### Quantum circuits for simulating weak decays

The initial state of our simulations is $\left|\psi_{init}\right\rangle = \left|\psi_{vac}^{(lep)}\right\rangle|\Delta^-\Delta^-\rangle$. Preparation of the lepton vacuum is straightforward, as it is two flavors of non-interacting fermions[120–122]. As mentioned above, the quark wavefunction factorizes between the $u$ and $d$ sectors as

$$|\Delta^-\Delta^-\rangle = \left|\psi_{vac}^{(u)}\right\rangle|0\rangle^{\otimes 6}. \tag{11}$$

The $|0\rangle^{\otimes 6}$ represents the fully occupied register of $d$ quarks and is trivial to prepare.

To prepare $\left|\psi_{vac}^{(u)}\right\rangle$, we use the Scalable Circuit ADAPT-VQE[123] (SC-ADAPT-VQE) algorithm developed in refs. 22,118. SC-ADAPT-VQE is a variational quantum algorithm that utilizes symmetries and hierarchies in length scales to determine shallow state preparation circuits. On a two spatial site lattice, the SC-ADAPT-VQE ansatz consists of a single parameterized circuit $e^{i\theta\widehat{O}}$ that is real and preserves the symmetries of the QCD Hamiltonian. An operator with these properties is

constructed from the commutator of the mass and kinetic terms (for the $u$ quarks) in $\widehat{H}_{free}$,

$$\widehat{O} = i \sum_{n=0}^{2L-1} \left[(-1)^n \phi_n^{(u)\dagger}\phi_n^{(u)}, \left(\phi_n^{(u)\dagger}\phi_{n+1}^{(u)} + \text{h.c.}\right)\right]. \tag{12}$$

The variational optimization and the construction of the associated quantum circuits is discussed at length in Supplementary Note 10.

Next, the decay process is simulated by time evolving the initial state with $e^{-iHt}$ using a first-order Trotterization. By judiciously ordering the terms in the Trotter decomposition, the first Trotter step can be simplified using $(\widehat{H}_{free} + \widehat{H}_{Maj} + \widehat{H}_{glue})\left|\psi_{init}\right\rangle = E_{init}\left|\psi_{init}\right\rangle$[79]. We introduce a new method for constructing the required time-evolution quantum circuits that builds on results presented in ref. 124. These circuits feature a high degree of parallelizability when compiled to a device with all-to-all connectivity. The circuit construction will briefly be described here, with more information in Supplementary Note 10. An obstacle in the way of highly parallelizable circuits is the JW transformation, which adds a string of Pauli-$\widehat{Z}$ operators to all non-mass terms in the Hamiltonian. This can be overcome by first designing the circuits without the JW $\widehat{Z}$ strings, and then including them by conjugating the $\widehat{Z}$-string-free circuits by a sequence of $CZ$s, taking advantage of the identity shown in (Fig. 5b).

For demonstration, consider the time-evolution of a kinetic term for one flavor of quark over one spatial site,

$$\widehat{H}_{kin} = \frac{1}{2}\sum_{c=0}^{2}\left[\widehat{\sigma}_c^+ \widehat{Z}_{c+1}\widehat{Z}_{c+2}\widehat{\sigma}_{c+3}^- + \text{h.c.}\right]. \tag{13}$$

The steps for creating a circuit that implements the unitary evolution of $e^{-it\widehat{H}_{kin}}$ are shown in (Fig. 5a). In the first equality, the fact that $(CZ)^2 = \widehat{I}$ (highlighted with the same color) and that $CZ$s commute with each other has been used. The second equality decomposes the orange box and pushes the $CZ$s to the beginning and end of the circuit. This strategy is used to construct all of the time-evolution circuits in this work. This circuit construction method generalizes ideas used in fermionic SWAP networks[122,125,126]. This is because the fermionic SWAP gate is equivalent to a qubit SWAP operation followed by a $CZ$ gate, the former of which can be implemented virtually with all-to-all connectivity.

## Data availability

The data that support the findings of this study are available from the corresponding author upon request.

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

## Acknowledgements

We would like to thank Saurabh Kadam, Joe Latone, and Torin Stetina for helpful discussions and support. This work was supported, in part, by U.S. Department of Energy, Office of Science, Office of Nuclear Physics, InQubator for Quantum Simulation (IQuS) under Award Number DOE (NP) Award DE-SC0020970 via the program on Quantum Horizons: QIS Research and Innovation for Nuclear Science (MJS, IC, RCF), and by the Quantum Science Center (QSC) which is a National Quantum Information Science Research Center of the U.S. Department of Energy (MI, IC). This work is also supported in part by Los Alamos National Laboratory, which is operated by Triad National Security, LLC, for the National Nuclear Security Administration of U.S. Department of Energy (Contract No. 89233218CNA000001) (IC). This work is also supported, in part, through the Department of Physics and the College of Arts and Sciences at the University of Washington. RCF acknowledges support from the U.S. Department of Energy QuantISED program through the theory consortium "Intersections of QIS and Theoretical Particle Physics" at Fermilab, from the U.S. Department of Energy, Office of Science, Accelerated Research in Quantum Computing, Quantum Utility through Advanced Computational Quantum Algorithms (QUACQ), and from the Institute for Quantum Information and Matter, an NSF Physics Frontiers Center (PHY-2317110). RCF additionally acknowledges support from a Burke Institute prize fellowship. This research used resources of the National Energy Research Scientific Computing Center (NERSC), a Department of Energy Office of Science User Facility using NERSC award NP-ERCAP0032083.

## Author contributions

A.A. designed and implemented the tool used in allocating choosing shot count per variant and number of variants; co-developed and co-implemented the bootstrapping tools used to estimate the error bars reported; analyzed and post-processed results from IonQ Forte and Enterprise; participated in bi-weekly meetings; co-wrote the manuscript. A.B. participated in bi-weekly meetings; co-wrote the manuscript. I.C. co-conceived the project and model Hamiltonian; co-developed the quantum circuits; performed classical simulations using exact diagonalization and statevector simulators; analyzed post-processed results from IonQ Forte and Enterprise; participated in bi-weekly meetings; co-wrote the manuscript. R.F. co-conceived the project and model Hamiltonian; co-developed the quantum circuits; performed classical simulations using exact diagonalization and statevector simulators; analyzed post-processed results from IonQ Forte and Enterprise; participated in bi-weekly meetings; co-wrote the manuscript. C.G. participated in bi-weekly meetings; co-wrote the manuscript. M.I. co-conceived the project and model Hamiltonian; co-developed the quantum circuits; performed classical simulations using exact diagonalization and statevector simulators; analyzed post-processed results from IonQ Forte and Enterprise; participated in bi-weekly meetings; co-wrote the manuscript. A.K. led the technical execution of the project and co-led the coordination of the activities of the IonQ and UW teams; executed the quantum circuits on the IonQ QPUs IonQ Forte and Enterprise machines; analyzed and post-processed results from hardware; participated in bi-weekly meetings; co-wrote the manuscript. M.A.L.-R. analyzed and post-processed the results from hardware runs; participated in bi-weekly meetings; co-wrote the manuscript. A.M. co-designed and co-conceived the error mitigation and error detection techniques for IonQ Forte and Enterprise machines; co-developed the quantum circuits; analyzed and post-processed results; participated in bi-weekly meetings; co-wrote the manuscript. M.R. co-led and co-conceived the project; analyzed post-

processed results from IonQ Forte and Enterprise; participated in bi-weekly meetings; co-wrote the manuscript. M.S. co-led and co-conceived the project; co-developed the model Hamiltonian; verified the quantum circuits; analyzed post-processed results from IonQ Forte and Enterprise; participated in bi-weekly meetings; co-wrote the manuscript. Y.deS. co-developed software infrastructure for error mitigation, compilation and circuit execution on IonQ Forte and Enterprise; analyzed and post-processed results from IonQ Forte and Enterprise; participated in bi-weekly meetings; co-wrote the manuscript. F.T. co-designed and co-developed the error mitigation and error detection techniques for IonQ Forte and Enterprise machines; co-developed software infrastructure for error mitigation, compilation, and circuit execution on IonQ Forte and Enterprise; co-developed the quantum circuits; analyzed and post-processed results; participated in bi-weekly meetings; co-wrote the manuscript.

## Competing interests

The authors declare the following competing interests: A.M., F.T., M.A.L.R., A.A., Y.D.S., A.B., C.G., A.K., and M.R. are employees and equity holders of IonQ, Inc. M.J.S. serves on an IonQ advisory committee and owns stock. I.A.C., R.C.F., and M.I. have no competing interests.
