## [Transparent Peer Review file · Nature Communications]

Pathfinding Quantum Simulations of Neutrinoless Double- β Decay

Corresponding Author: Dr Roland Farrell

Version 0:

Reviewer comments:

Reviewer #1

(Remarks to the Author)

This paper is an early attempt to model a nuclear neutrinoless double-beta decay using a simplified description of quarks and leptons in one spatial dimension, using a quantum computer.

The core result is using elements from previous papers to (co-)design a feasible experiment on a quantum computer with limited resources (qubits and gates fidelities)

The one dimensional spatial lattice contains only 2 sites, and several approximations are made, which are very important to contextualize the results, but are only discussed in the supplemental material.

I acknowledge that running these type of experiments requires some effort, as most quantum computing platforms have limited availability, but I do not recommend publication unless a big effort is made to rewrite the manuscript to highlight the novelties and, more importantly, the approximations that are made to obtain results.

The main issue is that the approximations listed in Appendix E.3 are very important to understand the main results in Fig. 3. Fig. 3 seems to imply that the mitigated results are showing the physics that one expects from exact diagonalization, therefore validating the quantum simulation. However, Fig. 10 tells a very different story. Namely, it is a real accident that the approximations used end up "close" to the exact diagonalization result. And it is probably a very lucky accident that the circuit requiring the least amount of gates is "close" to the correct result.

I think this should be emphasized instead of hidden away in Fig. 10.

When using less approximations and, as the authors stated, "pushing the limit" of the hardware, there is no signal, as shown in Fig. 4 (everything is statistically consistent with zero, meaning that the Majorana mass does nothing to the system).

What I take away from Fig. 3 is that the lepton number observable when there is no Majorana mass should be zero, regardless of the approximations made (e.g. it is zero also in Fig. 10) and error mitigation indeed recovers the result. However, we are talking about a local observable measured after only 500 two qubit gates, and this is not surprising at all on a trapped-ion device.

If the authors decide to re-write the content as I suggested, I would also request that a bit more effort is made in explaining the role of the "ad-hoc" parameters chosen in the Hamiltonian (Appendix C.) When reading the main text, a superficial reader might think the authors are talking about the QCD Hamiltonian with physical parameters, but in fact we are talking about a lattice Hamiltonian, on a crazily small lattice of 2 sites, with "fine-tuned" parameters to observe something that vaguely resembles neutrinoless double-beta decay in a "simple model of a nucleus".

To summarize, I am glad to see the effort in designing the experiments and I think the paper explains everything that was done in details. However, the main text lacks clarity when it comes to what the results actually mean and how they should be reported (approximations are crucial to understand why the points in Fig. 3 are where they are: an accident).

There is a small chance that I misunderstood Fig. 10, and that the very strong approximations made to reduce the Hamiltonian terms and the system degrees of freedom (leading to a reduction in number of gates) are actually naturally leading to the correct physics. If so, this would be a very important fact to highlight in the main text!

Reviewer #2

(Remarks to the Author)

Key Results:

The authors consider a quantum simulation model of double-beta decay with a tunable Majorana mass term and implement it on IonQ's Forte generation QPUs. After preparing a pair of baryons in the quark sector and the vacuum in the lepton sector, they evolve the system under a lattice Hamiltonian that includes a vector-like four-fermion coupling. The dibaryon initial state has six down quarks and at lowest order their weak decays only produce electrons and neutrinos with opposite lepton number. The rise of electric charge density in the leptonic sector is the primary signal of the double-beta decay. When the Majorana mass (m_M) is zero, this process conserves lepton number. When m_M is turned on, the evolution of electric charge is almost the same as in the massless case, but lepton number varies. This is taken to be a signal of neutrinoless double-beta decay.

Validity:

The reasonable matching of exact evolution, noiseless Trotter, and IonQ data illustrate the effectiveness of the state preparation, gate evolution, and error-mitigating post-selection.

Because of hardware limitation, the calculation involves only two sites. This is quite far from the ideal situation (infinite volume and continuum limits), where final states with two electrons and in addition 0, 1 or 2 neutrinos correspond unambiguously to the neutrinoless double beta decay, a single beta decay and the double beta decay with two neutrinos. With two sites, the authors monitor the electron and neutrino occupation and calculate the evolution of the charge and lepton number.

When $m_M=0$, the neutrino and electron occupation are the same.

When m_M is non zero, the neutrino occupation is suppressed.

If we understand correctly, the neutrino occupation is $-(Q+L)$ and can be mentally reconstructed from Fig. 3. However, it is not completely clear what is meant by neutrinoless.

There are only two choices of m_M (0 and 1,7)

and it is not completely clear why this number 1.7 was selected.

It might be interesting to have a better picture of the m_M dependence in the evolution of the neutrino occupation.

How does the lower right part of Fig. 3 depend on m_M ?

We believe that the ideal simulations would be suitable to give an idea about these questions.

Significance:

The form of the Hamiltonian is inspired by Standard Model interactions, but coaxing the model to prefer the desired decay channels requires an exotic choice of masses and couplings (Appendix C). The fact that practical challenges limit the possibility of realistic parameters is well-acknowledged in the conclusion. Despite these challenges, we believe that this work meaningfully demonstrates the value of compact, short-time scale simulation of decay processes.

Data and Methodology:

In Appendix H, three different error mitigation techniques are discussed, two of which involve post-selection of simulator data. It would be useful to know what fraction of the samples were excluded to get an idea of the efficiency of the simulator.

Analytical approach:

The basic tools for building the Hamiltonian and prepare the vacuum are discussed in published work by some of the authors and are reliable. The claim of 10 sigma effect in Fig. 3 seems plausible but some numerical detail may be useful.

Clarity and context:

It took some time to deduce that the 'n' index in eqns. 3-6 counted over staggered fermion sites. This is relevant to interpreting both 5 and 6 in terms of couplings between matter and anti-matter degrees of freedom. This could be made more clear within the body text of that section.

The article would fit well in Nature Communications with limited changes.

Suggested improvements:

- 1) Include the evolution of neutrino sector occupation number to show to what extent the observed decay is "neutrinoless".
- 2) Indicate the proportion of retained data after post-selection for error mitigation.

3) Clarify indices in Hamiltonian equations

4) Appendix A has a presentation that does not blend well with the main text where lepton number is described as conserved in the standard model ($m_M=0$). We believe that the breaking of B+L mentioned in Appendix A is due to finite temperature effects but it is not clear what it means in the context of the main text.

Minor suggestions (optional):

End of p. 1

Yocto-seconds are nine orders smaller than the femto seconds in chemistry but the size of the molecules is only 5 orders of magnitude larger. Would the energies involved be a better way to draw the analogy?

Beginning of p.2

"Stability of nuclei places tight constraint on structures beyond the standard model [35-40]."
The references [35-40] are the standard SM references and we believe have no discussion on nuclei stability.
Would additional references on nuclear stability be useful?

A bit below: "...coherent sum over low energy excitations. ..."
If this is more easily taken into account with quantum computing, "
Could real-time calculation be (in principle) converted into form factors used in the traditional approach (maybe using Fourier transform)?

General questions: what are the more stringent limits on $0\nu\beta\beta$ decay today, how would the new method improve these limits if the process is not observed directly?

Reviewer #3

(Remarks to the Author)

The manuscript presents a quantum simulation of a two-site lattice model for double beta decay using two advanced trapped-ion quantum processors. The experiment builds on previous single-site work [e.g., PRD 107, 054513 (2023)] and incorporates multiple co-design techniques, including post-selection, dynamical decoupling, and hardware-aware circuit design.

The overarching motivation to explore how quantum computing can support high-energy and nuclear physics is timely and well-justified. However, in its current form, the manuscript does not meet the clarity expected for Nature Communications. While the experimental scale and effort are certainly notable, the paper does not present a clear picture of what has been learned, what the limitations are, or why the results matter.

Several concerns:

1. There is a heavy reliance in the evolution on minimal Trotterization, often just a single step. Figures 3 and 9 show that this already introduces large deviations from ideal dynamics, in some cases changing the qualitative behavior. This is not a small error that can be ignored. This raises the question of how closely the simulated evolution reflects the intended Hamiltonian. Even for a two-site model and short evolution times, the discrepancy is so significant that it is hard to draw any physical conclusions from the output: For example, the lepton number in Fig. 9 varies drastically with the number of Trotter steps. If such results were interpreted as physical observables from a quantum device, they would correspond to many-sigma deviations from the true dynamics and would misrepresent the studied model. A more thorough discussion of both theoretical and experimental limitations of the taken approach is therefore essential.

2. The framing of the paper frequently overstates what has been achieved. References to "imaging sub-yoctosecond nuclear dynamics" suggest a direct physical probe that is not present in this work. The system evolves under a toy Hamiltonian using an approximate method. Claims about the "scalability of the platform" are also too strong given that no scaling analysis or roadmap is provided. Assertions about potential new physics applications are also too vague. These type of statements should be removed or rephrased to better reflect the demonstrated capabilities. The introductory paragraph- 'future simulations...' alludes to a physics experimental observation of lepton number violation tied to QCD model. While this paper only discusses a toy model with conservative approximations. A more targeted discussion that clearly frames the scope and relevance of the current quantum simulation experiment would help avoid potentially misleading interpretations and overstatements of significance.

3. The conceptual contribution of the work is unclear. If the goal is to benchmark the readiness of quantum hardware for high-energy simulations, that should be stated directly. The paper should then critically examine what the key bottlenecks are, which mitigation techniques contributed significantly, and whether the approach is extensible to more realistic models. For example, Fig. 4 involves a more complex circuit on a different processor, but the experimental errors are significantly larger

and it is not obvious what additional insight is gained compared to Fig. 3. Without this kind of discussion, the manuscript remains a technical demonstration rather than a scientific study. In that spirit, it is also hard to judge which of the co-designed techniques are "fine-tuned" to the hardware, and how much of that fine-tuning can be potentially maintained at larger scale or models. This includes for example the VQE stage for state preparation and application of post selection.

4. The presentation also needs significant improvement. For example, section 2 discusses post-selection before the model is even defined. Terms like "twirled variants", "JW Z string" are used without explanation. The overall procedure, including state preparation, Trotterized evolution, and (mid-circuit/final) measurement can be shown schematically. Reorganization into a more conventional structure with Introduction, Results, and Discussion would help improve clarity and accessibility. The authors should also improve the discussion of Fig. 1, both in the text and caption, clearly distinguishing differences between L, I, and physical qubits. Currently they repeat the same spatial site indices (I), which can be confusing.

5. The authors rely heavily on post-processing and noise simulations due to low survival rate of the experimental shots. They use it for circuit design where they assign qubits to de-bias qubit associated error dependencies, and post select bit strings to conserve color and total charge. However, a more intuitive and transparent discussion of how these techniques influence the physical observables and overall interpretation of results is missing. Furthermore, in the discussion of the result the authors claim that "debiasing with nonlinear filtering is a superior error mitigation strategy" is made without sufficient justification or comparative analysis. In the discussion of Fig. 3, the authors provide little insight into the choice of the Majorana mass m_M values. In the same figure caption, the authors should clearly indicate the relevant parameters of the experimental data (i.e. trotter steps) and the simulations (approximations).

In its current form, the manuscript is not suitable for Nature Communications. That said, we do believe the work has strong foundations and could become more suitable for this venue if the authors revise it substantially. A more focused and critically reflective manuscript: one that better clarifies the scope and limitations of the theoretical and experimental approach, avoids overstated claims, but highlights the actual contributions and offers insight into what future experiments or improvements are needed. This could provide real value to the quantum simulation and HEP communities. As it stands, the work may be more appropriate for a focused journal such as Communications Physics and can be recommended for publication there with minimal revisions.

Reviewer #4

(Remarks to the Author)

Reviewer #5

(Remarks to the Author)

Version 1:

Reviewer comments:

Reviewer #1

(Remarks to the Author)

The authors have taken my criticism very seriously and have worked hard to change the structure of the paper to address them.

In its current form the paper should be published without further review on my side.

However, I want to point out that the added paragraph in the discussion section about the IonQ trapped-ion roadmap seems a bit out of place, because it is not connected to the rest of the paper. For example, if the added paragraph talks about 800 logical qubits with $1e-7$ error rate in 2027, I would also suggest to add what kind neutrinoless double beta decay lattice model could be achieved with it.

Reviewer #2

(Remarks to the Author)

The three referees concurred to ask for details regarding approximations and fine-tunings and to make sure that existing limitations are stated more clearly earlier in the main text.

The authors have addressed their concerns carefully in the reply and in the text, in a way that I consider satisfactory. The significance of the work as well as its limitations is clarified. The new figures complement the revised text efficiently.

An impressive set of state-of-the-art techniques for quantum computing has been used. It is well documented and will be a standard reference to assess the progress in this direction in the coming years. The approximations and fine-tunings put the

possible observation of lepton number violation in perspective. The possible existence of a Majorana mass term is a central question in neutrino physics.

The current manuscript meets the criteria of Nature Communications. It represents a first step towards a quantum computation of the neutrinoless beta decay in actual nuclei and it will be a standard reference in the coming years.

Reviewer #3

(Remarks to the Author)

We appreciate the authors' clarifications and do not object to the use of a simplified or reduced-dimensional Hamiltonian. The core issue, however, is not the model itself but the number of approximations (e.g. low-order Trotterization, weak-operator truncation, unphysical parameter choices, and significant post-selection), whose combined effect is not quantitatively controlled. These approximations are then used to underpin claims about realization of meaningful results in 2+1D and 3+1D, along with scalability to larger systems- that, in our view, cannot be justified without the systematic analysis that is missing in the paper. Moreover, the manuscript's own results suggest that the Trotterized evolution is only reliable at very short times (unless many steps are applied), which further limits the scope of the conclusions. Overall, while the work has merit as a proof-of-principle demonstration, we do not believe it meets the standards of Nature Communications, and we would consider it more appropriate for a venue such as Communications Physics.

Reviewer #5

(Remarks to the Author)

We would like to thank the reviewers for their careful reading of our manuscript, as well as the detailed feedback. Here we address the points raised by the referees. In the new version of the manuscript, we have marked the corresponding changes in red.

Note that in the latest version of the manuscript, Fig. 3 is now Fig. 5, Table 1 is now Table 2, Table 2 is now Table 1, Fig. 10 is now Fig. 11, Appendix B2 is now Appendix B3, and Appendix E is now Appendix F.

Referee 1:

This paper is an early attempt to model a nuclear neutrinoless double-beta decay using a simplified description of quarks and leptons in one spatial dimension, using a quantum computer.

The core result is using elements from previous papers to (co-)design a feasible experiment on a quantum computer with limited resources (qubits and gates fidelities) The one dimensional spatial lattice contains only 2 sites, and several approximations are made, which are very important to contextualize the results, but are only discussed in the supplemental material.

I acknowledge that running these type of experiments requires some effort, as most quantum computing platforms have limited availability, but I do not recommend publication unless a big effort is made to rewrite the manuscript to highlight the novelties and, more importantly, the approximations that are made to obtain results.

The main issue is that the approximations listed in Appendix E.3 are very important to understand the main results in Fig. 3.

Fig. 3 seems to imply that the mitigated results are showing the physics that one expects from exact diagonalization, therefore validating the quantum simulation.

(i) Our intent of plotting the “exact” curve in Fig. 3 (now 5) was not to show agreement with the results from the device, but was shown for comparison. We acknowledge that it was potentially misleading and could be interpreted as downplaying the approximation errors. To avoid this confusion, we have removed the exact line from Fig. 3 (now 5) and Fig. 4. Also, we have unified the style of the lines in Fig. 3 (now 5) and Fig. 4 representing the noiseless results with those of Fig. 10 (now 11). This makes it easier to identify which approximations were employed in Fig. 3 (now 5) and Fig. 4.

However, Fig. 10 tells a very different story. Namely, it is a real accident that the approximations used end up “close” to the exact diagonalization result. And it is probably a very lucky accident that the circuit requiring the least amount of gates is “close” to the correct result. I think this should be emphasized instead of hidden away in Fig. 10.

(ii) Indeed, it is an accident that the employed approximation provides a curve that is close to the exact result over the considered time interval, and with the change described above, we hope that this is clear and ambiguity free. For further clarifications, we have:

- Modified the content of Fig. 10 (now 11) to remove two of the intermediate truncations that we considered but did not implement.
- Modified the captions of Fig. 3 (now 5) and Fig. 4 to specify some of the approximations that were made.
- Added a sentence to the first paragraph of Sec. V.A:
The effects of these approximations are detailed in Appendix F. Despite the errors due to approximation becoming significant for $t \geq 1.0$, our simulations are still able to extract qualitatively correct signals of $0\nu\beta\beta$ -decay.
- Added Appendix B2 providing details on the four-Fermi weak interaction that was used and the approximations that were made: *The weak interactions giving rise to single- β decay are modeled through a local vector-like*

four-Fermi operator [1],

$$\begin{aligned}
\hat{H}_\beta^{1+1} &= \frac{G}{\sqrt{2}} \int d^2x (\bar{\psi}_u \gamma^\mu \psi_d \bar{\psi}_e \gamma_\mu \mathcal{C} \psi_\nu + \text{h.c.}) \\
&\rightarrow \frac{G}{\sqrt{2}} \sum_{n \text{ even}} \left[\left(\phi_n^{(u)\dagger} \phi_n^{(d)} + \phi_{n+1}^{(u)\dagger} \phi_{n+1}^{(d)} \right) \left(\chi_n^{(e)\dagger} \chi_{n+1}^{(\nu)} - \chi_{n+1}^{(e)\dagger} \chi_n^{(\nu)} \right) \right. \\
&\quad \left. + \left(\phi_n^{(u)\dagger} \phi_{n+1}^{(d)} + \phi_{n+1}^{(u)\dagger} \phi_n^{(d)} \right) \left(\chi_n^{(e)\dagger} \chi_n^{(\nu)} - \chi_{n+1}^{(e)\dagger} \chi_{n+1}^{(\nu)} \right) + \text{h.c.} \right] \\
&\approx \frac{G}{\sqrt{2}} \sum_{n \text{ even}} \left(\phi_n^{(u)\dagger} \phi_n^{(d)} \phi_n^{(e)\dagger} \phi_{n+1}^{(\nu)} + \text{h.c.} \right), \tag{1}
\end{aligned}$$

where \mathcal{C} is the charge-conjugation operator, γ^μ are the gamma-matrices and G is the weak coupling constant (Fermi's constant). The first line shows the 1+1D interaction is related to the low-energy charged-current weak interaction of the Standard Model. The mapping of the fermion fields to a staggered lattice is shown in the second line. This necessarily includes contribution from both particles and anti-particles due to operator contractions. The results displayed in Fig. 4 are obtained using this interaction. The third line employs a ‘‘valence-fermion’’ approximation that only keeps the terms acting on the valence-quarks (no operators acting on anti-quark sites) and valence-leptons (no operators acting on neutrino or anti-electron sites). The results displayed in Fig. 5 are obtained using this approximation.

- Added a paragraph to Appendix B3 (previously B2), explicitly presenting the valence weak interaction, and with further discussion: *After the JW mapping, the complete charged-current weak interaction (modeled in 1+1D) is given in Eq. (B8d). It includes operators acting on both the valence- and sea-fermions, i.e., an up-quark operator can create an up-quark or it can annihilate an anti-up-quark, both actions increase the up-quark number by one. For a complete simulation, all possible such actions should be included. However, retaining only the contributions from the operators acting on valence-fermions is a well-defined approximation that can be implemented, reducing the required depth of quantum circuits. Equation (B11) gives the Hamiltonian describing only the valence weak interactions, the approximation that we make in obtaining the results shown in Fig. 5 and shown in Table II,*

$$\hat{H}_{\beta, \text{valence}}^{1+1} = \frac{G}{\sqrt{2}} \sum_{l=0}^1 \sum_{c=0}^2 \hat{\sigma}_{24+4l+2}^- \hat{\sigma}_{24+4l+1}^+ \hat{\sigma}_{12l+3+c}^- \hat{Z}^2 \hat{\sigma}_{12l+c}^+ .$$

When using less approximations and, as the authors stated, ‘‘pushing the limit’’ of the hardware, there is no signal, as shown in Fig. 4 (everything is statistically consistent with zero, meaning that the Majorana mass does nothing to the system).

(iii) The referee is correct that the noisy results are statistically consistent with zero for the lepton electric charge and are only one standard deviation (1σ) above zero for the lepton number, however we emphasize that it is the noise from the device that renders the effect of the Majorana mass indistinguishable from zero (and not that the Majorana mass term does not effect the system). As mentioned above, we have removed the ‘‘exact’’ line from Fig. 4, and modified the style of the line of the noiseless simulation (as in Fig. 10, now 11) to highlight the fact that it involves different approximations compared to Fig. 3 (now 5).

The reason we included Fig. 4 in the main text was not to suggest that we had obtained precise results from these deeper circuits, but to highlight an important part of the co-design process. The results shown in Fig. 4 were the first set of circuits that we ran on Forte, and were crucial for guiding our choice of approximations and error mitigation strategies used in subsequent runs on Forte Enterprise. For example, we discovered that leakage detecting flags were more effective than mid-circuit symmetry checking flags (via controlled iSWAP operations). The acute signs of device decoherence obtained from these deeper circuits also led us to consider additional approximations (as discussed in Appendix F). This ultimately led us to the simplified simulations presented in Fig. 3 (now 5) that were still able to capture the relevant physics of lepton number violation induced by neutrinoless beta decay. To make this clearer, we have added additional text to section V.B to address this point.

Further context about the quantum computers used in this work has been added to Section II.B, clarifying that the two different devices used, Forte and Forte Enterprise, have the same design. Forte was manufactured first and Forte Enterprise was manufactured second. It is because of their similarity that we were able to directly apply the lessons learned from the first experiments on Forte to the second set of experiments run on Forte Enterprise. This iteration was a crucial component of the co-design process.

What I take away from Fig. 3 is that the lepton number observable when there is no Majorana mass should be zero, regardless of the approximations made (e.g. it is zero also in Fig. 10) and error mitigation indeed recovers the result. However, we are talking about a local observable measured after only 500 two qubit gates, and this is not surprising at all on a trapped-ion device.

(iv) The referee is interpreting correctly the results in the middle column of Fig. 3 (now 5) and Fig. 10 (now 11). An appealing feature of this simulation is that we are able to extract interesting physics results from such simple observables (local observables). In principle, we could also study non-local observables (correlators, entanglement structure), but have not done so in this initial work. Prior to our work, the number of gates required to implement any approximations for $0\nu\beta\beta$ -decay was unknown. The initial set of experiments on Forte, with fewer approximations, established the current limit of quantum simulation for this model system. We now understand what is currently possible for these systems, and what is required for implementation of the weak operator with both sea- and valence-fermions. While the current results (obtaining a signal for a local observable from a ~ 500 entangling-gate circuits) should not be a surprise to experts, this work provides the first demonstrations of such, and identifies the implications for the targeted systems. Additionally, determining unbiased estimates of observables with $\sim 5\%$ -level precision (e.g. $\langle \hat{Q}_e \rangle_{m_M=1.7}$ at $t = 2$ in Table 1 (now 2)) from a circuit with 500 entangling gates (each with 99.5% fidelity) is noteworthy. This highlights the utility of the error mitigation methods that were used and the power of Forte Enterprise.

If the authors decide to re-write the content as I suggested, I would also request that a bit more effort is made in explaining the role of the "ad-hoc" parameters chosen in the Hamiltonian (Appendix C). When reading the main text, a superficial reader might think the authors are talking about the QCD Hamiltonian with physical parameters, but in fact we are talking about a lattice Hamiltonian, on a crazily small lattice of 2 sites, with "fine-tuned" parameters to observe something that vaguely resembles neutrinoless double-beta decay in a "simple model of a nucleus".

(v) The parameters of the Hamiltonian were deliberately tuned to reproduce the mass hierarchy that naturally occurs in certain nuclei such as ^{76}Ge , and allows $\beta\beta$ -decay to be kinematically accessible (requiring that β -decay is forbidden but $\beta\beta$ -decay allowed, made possible by the structure of the low-energy nuclear forces). Of course the Hamiltonian parameters chosen for our 1+1D simulations do not resemble the parameters in the QCD Hamiltonian for many reasons, including those mentioned by the referee. The following modifications have been implemented to address these concerns:

- As mentioned in the summary and conclusions section, for future simulations with larger systems, a selection of parameters (still away from their physical values) will be used to perform an extrapolation to the physical point.
- Words were added to the abstract: *Enabled by tuned model parameters, a clear signal of neutrinoless double- β decay is measured...*
- Sentences were added to the Introduction: *Specifically, we perform lattice simulations of the decay of two baryons restricted to two spatial sites. Both strong and weak interactions are included, and a Majorana neutrino mass term (that explicitly violates lepton-number conservation). The coupling constants and masses are deliberately tuned to recover a mass hierarchy that kinematically favors double- β decay, but suppresses single- β decay (in this volume).*
- The values of the Hamiltonian parameters have been added to the main text (Sec. III), together with the following clarifying footnote: *We emphasize that the parameters defining the 1+1D Hamiltonian we have implemented are not related to those in nature. They are selected to permit a model simulation of double- β decay that is a first step toward future simulations in this genre of fundamental physics. Appendix D provides a short discussion of future extrapolations that will be required to be able to make predictions for physical observables.*

To summarize, I am glad to see the effort in designing the experiments and I think the paper explains everything that was done in details. However, the main text lacks clarity when it comes to what the results actually mean and how they should be reported (approximations are crucial to understand why the points in Fig. 3 are where they are an accident).

(vi) We again thank the referee for their positive comments and critiques. We hope that we have addressed these comments and concerns, which we believe have improved the quality of our manuscript.

There is a small chance that I misunderstood Fig. 10, and that the very strong approximations made to reduce the Hamiltonian terms and the system degrees of freedom (leading to a reduction in number of gates) are actually naturally leading to the correct physics. If so, this would be a very important fact to highlight in the main text!

(vii) The referee has not misunderstood Fig. 10 (now 11), and we hope to have removed ambiguity in this regard, see reply (ii).

Referee 2:

Key Results:

The authors consider a quantum simulation model of double-beta decay with a tunable Majorana mass term and implement it on IonQ's Forte generation QPUs. After preparing a pair of baryons in the quark sector and the vacuum in the lepton sector, they evolve the system under a lattice Hamiltonian that includes a vector-like four-fermion coupling. The dibaryon initial state has six down quarks and at lowest order their weak decays only produce electrons and neutrinos with opposite lepton number. The rise of electric charge density in the leptonic sector is the primary signal of the double-beta decay. When the Majorana mass (m_M) is zero, this process conserves lepton number. When m_M is turned on, the evolution of electric charge is almost the same as in the massless case, but lepton number varies. This is taken to be a signal of neutrinoless double-beta decay.

Validity:

The reasonable matching of exact evolution, noiseless Trotter, and IonQ data illustrate the effectiveness of the state preparation, gate evolution, and error-mitigating post-selection.

(viii) In light of the previous referee's comments, we want to clarify a point about the agreement between the exact evolution, noiseless simulation and IonQ Forte data. The approximations employed in Fig. 3 (now 5) and Fig. 4 are quite significant, and the agreement between, for example, the noiseless and exact results are mostly due to an accidental cancellation of Trotter and approximate-interaction errors. This is made clear in Fig. 10 (now 11) which shows the effects that each level of approximation has on the observables. To avoid potential confusion, we have removed the line representing the exact evolution from Fig. 3 (now 5) and Fig. 4, and added a number of clarifying statements to make this clearer - see replies (i)-(ii)

Because of hardware limitation, the calculation involves only two sites. This is quite far from the ideal situation (infinite volume and continuum limits), where final states with two electrons and in addition 0, 1 or 2 neutrinos correspond unambiguously to the neutrinoless double beta decay, a single beta decay and the double beta decay with two neutrinos. With two sites, the authors monitor the electron and neutrino occupation and calculate the evolution of the charge and lepton number. When $m_M = 0$, the neutrino and electron occupation are the same. When m_M is non zero, the neutrino occupation is suppressed. If we understand correctly, the neutrino occupation is $-(Q + L)$ and can be mentally reconstructed from Fig. 3. However, it is not completely clear what is meant by neutrinoless.

(ix) As the referee correctly points out, finite-volume effects are significant in our simulations, thus inhibiting the direct observation of a decay with no neutrinos in the final state. All that can be claimed in our simulations is that turning on the Majorana mass shifts the number of neutrinos (N_ν) toward zero while not suppressing the number of electrons, and thus there are lepton-number violating decays. For example, one such contributing process that is allowed is a single beta decay followed by an application of the Majorana mass term taking, e.g., $\Delta^-\Delta^- \rightarrow \Delta^0\Delta^- + e^- + \nu$. Although this process is disfavored kinematically due to our deliberate tuning of the Hamiltonian parameters (see reply (v) above), it will still occur due to finite-size effects. Furthermore, its final state has $N_\nu = 1$, and would contribute to moving the expectation value, $\langle N_\nu \rangle$, toward zero.

To clarify this point, we have modified Sec. V.C and added new panels to Figs. 3 (now 5), 4, 9 (now 10) and 10 (now 11), where we show the number of neutrinos

$$\hat{N}_\nu = \# \text{ of neutrinos} - \# \text{ of anti-neutrinos} = \hat{Q}_e + \hat{\mathcal{L}} \quad (2)$$

as a function of time. For $m_M = 0$, this quantity is purely negative, indicating the presence of anti-neutrinos in the final state. For $m_M = 1.7$, this quantity is still negative, but greatly reduced in absolute value. A particularly striking example is at $t = 2$ in Fig. 3 (now 5) where N_ν is almost zero while Q_e is large and negative. This can only be the case when both neutrinoless and neutrinoless decays are occurring in superposition. We note that simulations using this Hamiltonian in larger spatial volumes (when quantum computers become sufficiently capable) are expected to exhibit a further suppression of the decay channels that are kinematically disfavored.

There are only two choices of m_M (0 and 1.7) and it is not completely clear why this number 1.7 was selected. It might be interesting to have a better picture of the m_M dependence in the evolution of the neutrino occupation. How does the lower right part of Fig. 3 depend on m_M ? We believe that the ideal simulations would be suitable to give an idea about these questions.

(x) Similar to the comment from referee 1, see reply (v), we selected these parameters to reproduce a mass hierarchy that is similar to that found in nature, for this particular 1+1D simulation on two spatial sites. While studying the dependence of the observables on m_M is interesting, it would spoil the mass hierarchy engineered for this small system, requiring further fine tuning. As larger systems become possible to simulate, this restriction of parameters will relax, and a larger range can be explored. Of course, impactful simulations can still be performed with unphysical parameters, particularly in the neutrino sector. Effective field theory and other analytic results are expected to provide reliable ways to extrapolate from unphysical to physical parameters, thereby providing robust predictions.

Significance:

The form of the Hamiltonian is inspired by Standard Model interactions, but coaxing the model to prefer the desired decay channels requires an exotic choice of masses and couplings (Appendix C). The fact that practical challenges limit the possibility of realistic parameters is well-acknowledged in the conclusion. Despite these challenges, we believe that this work meaningfully demonstrates the value of compact, short-time scale simulation of decay processes.

Data and Methodology:

In Appendix H, three different error mitigation techniques are discussed, two of which involve post-selection of simulator data. It would be useful to know what fraction of the samples were excluded to get an idea of the efficiency of the simulator.

(xi) As stated in Sec. V.C, $\sim 10\%$ of the measurements survived post-selection for the quantum simulations presented in Fig. 3 (now 5).

Analytical approach:

The basic tools for building the Hamiltonian and prepare the vacuum are discussed in published work by some of the authors and are reliable. The claim of 10 sigma effect in Fig. 3 seems plausible but some numerical detail may be useful.

(xii) This $10\text{-}\sigma$ statement comes from examining the results in Table 1 (now 2). Specifically, the difference between the measured lepton numbers for $m_M = 1.7$ and $m_M = 0$ at $t = 2.0$.

Clarity and context:

It took some time to deduce that the ‘n’ index in eqns. 3-6 counted over staggered fermion sites. This is relevant to interpreting both 5 and 6 in terms of couplings between matter and anti-matter degrees of freedom. This could be made more clear within the body text of that section.

(xiii) We understand that the indexing is “heavy” in some places. Here are the modifications we have done to address this issue:

- A sentence has been added in Sec. III after Eq. (2): *The index n labels the staggered site, with n even corresponding to fermion sites (quarks and leptons), and n odd to anti-fermion sites (anti-quarks and anti-leptons).*
- We have modified Sec. III to only reference staggered sites.
- We have added Eq. (B11) and a surrounding paragraph, to clarify the Jordan-Wigner implementation of the valence fermion weak operator.

The article would fit well in Nature Communications with limited changes.

Suggested improvements:

1) *Include the evolution of neutrino sector occupation number to show to what extent the observed decay is “neutrinoless”.*

(xiv) We have included this new observable, as mentioned above in (ix).

2) *Indicate the proportion of retained data after post-selection for error mitigation.*

(xv) We have included these numbers, as mentioned above in (xi).

3) *Clarify indices in Hamiltonian equations*

(xvi) Appendix B2 has been added, elaborating the discussion regarding the weak interactions, and Appendix B3 has been expanded. The text around Eq. (2) and Eq. (5) has been modified, with a discussion of the indices - see (xiii).

4) *Appendix A has a presentation that does not blend well with the main text where lepton number is described as conserved in the standard model ($m_M = 0$). We believe that the breaking of $B+L$ mentioned in Appendix A is due to finite temperature effects but it is not clear what it means in the context of the main text.*

(xvii) We have removed the paragraph related to $B - L$ conservation and $B + L$ violation in spheralon processes and electroweak baryogenesis from Appendix A. A footnote has been added to the introduction clarifying accidental vs exact symmetries.

The new footnote text is: *Lepton number is an accidental classical symmetry of the Standard Model that is broken by quantum fluctuations.*

Minor suggestions (optional):

End of p. 1:

Yocto-seconds are nine order smaller than the femto seconds in chemistry but the size of the molecules is only 5 order of magnitude larger. Would the energies involved be a better way to draw the analogy?

(xviii) We have included a sentence to the second paragraph of the introduction discussing the lifetime of the Δ -resonance, justifying the use of yocto-seconds as characterizing the time scale over which quarks and gluons reorganize themselves at the hadronic scale. The hierarchy of length scales, permitting the well-established effective field theory description of nuclear forces, means that there are a set of well-defined time-scales for processes to take place.

The new text is: *This is the time scale relevant to hadronic structure with, e.g., the Δ -resonance decaying to a proton with a half-life $\tau_\Delta \sim 5$ ys.*

Beginning of p.2

“Stability of nuclei places tight constraint on structures beyond the standard model [35-40]. ” The references [35-40] are the standard SM references and we believe have no discussion on nuclei stability. Would additional references on nuclear stability be useful?

(xix) We agree with the referee, and we have added: “ [...] obtained from proton-decay, neutron-antineutron-oscillations, and the β -decay and $\beta\beta$ -decay of nuclei, see, e.g., Refs. [2-7]”. This was an oversight on our part.

A bit below: “...coherent sum over low energy excitations... If this is more easily taken into account with quantum computing,” Could real-time calculation be (in principle) converted into form factors used in the traditional approach (maybe using Fourier transform)?

(xx) This is an interesting point raised by the referee. For a system of non-interacting particles, such an approach would be possible, with the double-weak form factor of two-nucleon time-ordered products. Even for dilute systems, this might work. However, for strongly-correlated systems, such as a nucleus, there are contributions to the doubly-weak form factor from two single-weak form factors, and also from a tower of contact operators that recover the effect of two- and higher-nucleon correlations. These arise from the compositeness of the nucleons, with a size set by the chiral symmetry breaking scale. The lattice QCD work in Euclidean spacetime, such as Ref. [8] implements such a form factor type calculation for the long-distance contributions, expanding the amplitude for the decay in terms of as sum over matrix elements of the charged-current over all possible intermediate states (assuming effectively massless neutrinos). This method has limitations when transforming from Euclidean to Minkowski space, in ways analogous to the Maiani-Testa theorem for scattering. References to these works are now included in the Introduction (as well as in an appendix). At this time, we do not see a method that is obviously more efficient computationally than what we have implemented (for real-time dynamics). However, a combined use of Euclidean-space lattice QCD simulations of the long-distance contributions, matched to low-energy effective field theory, may provide complementary results

to future quantum simulations. Comments have been added to the introduction and conclusions addressing these points.

General questions: what are the more stringent limits on $0\nu\beta\beta$ -decay today, how would the new method improve these limits if the process is not observed directly?

(xxi) A nice summary of the present experimental constraints resulting from searches for $0\nu\beta\beta$ -decay can be found in the 2024 review from the Particle Data Group [4], and a representative set of references to current theoretical progress is given in the Introduction [9–15]. The issue we have addressed is the computation of the process, highlighting the use of quantum computers to compute the relevant nuclear matrix element. This provides a real-time calculation of the process, which is beyond what is possible with all other methods. In real-time simulations with finite spatial volumes and temporal evolution, the lifetime is extracted from the time-dependent change of lepton number, which will only be approximately exponential over some time interval. This added complication is now described in Appendix-D of the manuscript, with added citation to Ref. [1]. Classical computing and analytic techniques have significantly reduced the uncertainty in these matrix elements, however, even today, the uncertainties are significant. If this process is observed experimentally, or upper limits are improved, the constraint that such results would place on fundamental physics has uncertainties because of this. The correlations and short-range interactions in nuclei, from the strong interactions between nucleons, is the cause of these computational difficulties.

Referee 3:

The manuscript presents a quantum simulation of a two-site lattice model for double beta decay using two advanced trapped-ion quantum processors. The experiment builds on previous single-site work [e.g., PRD 107, 054513 (2023)] and incorporates multiple co-design techniques, including post-selection, dynamical decoupling, and hardware-aware circuit design.

The overarching motivation to explore how quantum computing can support high-energy and nuclear physics is timely and well-justified. However, in its current form, the manuscript does not meet the clarity expected for Nature Communications. While the experimental scale and effort are certainly notable, the paper does not present a clear picture of what has been learned, what the limitations are, or why the results matter.

Several concerns:

1. There is a heavy reliance in the evolution on minimal Trotterization, often just a single step. Figures 3 and 9 show that this already introduces large deviations from ideal dynamics, in some cases changing the qualitative behavior. This is not a small error that can be ignored. This raises the question of how closely the simulated evolution reflects the intended Hamiltonian. Even for a two-site model and short evolution times, the discrepancy is so significant that it is hard to draw any physical conclusions from the output: For example, the lepton number in Fig. 9 varies drastically with the number of Trotter steps. If such results were interpreted as physical observables from a quantum device, they would correspond to many-sigma deviations from the true dynamics and would misrepresent the studied model. A more thorough discussion of both theoretical and experimental limitations of the taken approach is therefore essential.

(xxii) We would like to clarify a couple of points regarding the Trotter errors. The simulations that were performed on IonQ’s quantum computers used $n_T = 2$ Trotter steps to access total simulation times $t = \{0.5, 1.0, 1.5, 2.0\}$ (e.g. see Fig. 3 (now 5)). The corresponding Trotter errors are illustrated in Fig. 9 (now 10) which shows the convergence of observables with n_T . We note that while the analysis in Fig. 9 (now 10) extends out to $t = 6$, the quantum simulations that we perform are only up to $t = 2$. For $t \leq 2$, the $n_T = 2$ curves in Fig. 9 (now 10) are close to the exact curves. In fact, it is seen in Fig. 9 (now 10) that the approximate valence-fermion weak interaction that we have used is a larger source of error than those due to Trotterized time evolution. Indeed, the referee is correct in saying that the cumulative effect of our approximation errors are not negligible, and are important to contextualizing our quantum simulation results. To address this, we have clarified the effects of these approximations in our responses (i) and (ii) to referee 1.

2. The framing of the paper frequently overstates what has been achieved. References to “imaging sub-yoctosecond nuclear dynamics” suggest a direct physical probe that is not present in this work. The system evolves under a toy Hamiltonian using an approximate method. Claims about the “scalability of the platform” are also too strong given that no scaling analysis or roadmap is provided.

(xxiii) The reference to imaging nuclear dynamics is intended to highlight a future potential application of quantum computers for nuclear physics problems, and not a result of the current manuscript. We have added a couple of clarifying words to the second paragraph of the introduction addressing this.

The Hamiltonian used in this work is not a “toy” Hamiltonian in the sense of being an artificial and simple Hamiltonian with no direct connection to QCD. The Hamiltonian employed in our work is a 1+1D version of the full 3+1D QCD plus weak interaction Hamiltonian, with the same symmetries (but which does have the limitation of not having dynamical gauge degrees of freedom). The approximations we have used are motivated by certain phenomenological features of 1+1D QCD that are also present in 3+1D QCD. For example, our truncation on the range of chromoelectric interactions in Eq. (3) is motivated by the exponential decay of correlations beyond the confinement scale [16]. Additionally, our usage of an approximate valence-fermion weak interaction is in the spirit of quenched simulations in Euclidean lattice QCD that omit dynamics of the sea quarks. Footnote 5 has been added to Sec. III making this connection: *Similar operator truncations were used in early quenched lattice QCD calculations, for example Ref. [17].* All of these approximations can be systematically improved with the availability of more powerful quantum computers. While it is the case that the parameters defining the Hamiltonian are not those of nature, this is generally true for any first simulation of a previously unexplored physical process. For example, lattice QCD worked with unphysical parameters (and excluded non-valence quarks for decades), and still does in many instances, with physical predictions obtained from extrapolations and interpolations to the physical point. We have added a discussion in Appendix D to highlight the historical importance of simulations away from the physical point in Euclidean lattice QCD, and their expected analogous role in quantum simulations. The changes outlined in response (v) to referee-1 have also been made to clarify our usage of unphysical parameters.

Regarding the scalability of the platform, we have added a final paragraph to Sec. VI that discusses the the most recent publicly available IonQ hardware development Road Map, and it’s implications for the studied problem. Additionally, we have included more analysis on the scalability of our error-mitigation strategy using debiasing with non-linear filtering in Appendix G.

Assertions about potential new physics applications are also too vague. These type of statements should be removed or rephrased to better reflect the demonstrated capabilities. The introductory paragraph- ‘future simulations...’ alludes to an physics experimental observation of lepton number violation tied to QCD model. While this paper only discusses a toy model with conservative approximations. A more targeted discussion that clearly frames the scope and relevance of the current quantum simulation experiment would help avoid potentially misleading interpretations and overstatements of significance.

(xxiv) The paragraph in the introduction that the referee is commenting on is important for emphasizing why the path established by our simulations may eventually impact experimental searches for BSM physics. These sentences are accurate and belong in the introduction. It is widely believed that quantum simulations of physical processes (using QCD and effective weak interactions) enabled by quantum computers will, in the future, aid in the (potential) discovery of new physics, enable improved limits to be placed on new physics, and provide predictions of emergent phenomena from known physics that are not possible using classical computing. Lattice QCD has already shown that classical simulations of QCD with effective weak interactions provide the highest precision strong interaction inputs required to extract/constrain new physics from experimental observations of certain systems/processes. Quantum computers will extend this into the domain where real-time dynamics are important. We have added additional sentences to the fifth paragraph of the introduction “The purpose of the present work...” to emphasize the path finding nature of our work. Additionally, the changes mentioned in response (v) to referee-1 have been implemented to make clear the approximations we have employed.

3.The conceptual contribution of the work is unclear. If the goal is to benchmark the readiness of quantum hardware for high-energy simulations, that should be stated directly. The paper should then critically examine what the key bottlenecks are, which mitigation techniques contributed significantly, and whether the approach is extensible to more realistic models.

(xxv) Upon re-reading the manuscript, we are inclined to agree with the referee that we did not clearly articulate the overarching objectives of this work. We have added the sentences beginning *The purpose of the present work is...* to the Introduction to rectify that situation. Along with the other changes, we hope that they are now clear.

With regard to more realistic models, we have made comments in reply (xxiii) related to this matter above. The Hamiltonian we are using is not arbitrary, it is the correct form of the Standard Model at low-energies in 1+1D, but simulated in a small volume with unphysical parameters. As all of these differences from the target simulations are controllable, we understand how to systematically remove them - in principle. As in lattice QCD simulations, where

exactly the same truncations and approximations are handled (modulo the different number of spatial dimensions), the path forward to remove them and connect with nature rests in the capabilities of quantum computers and associated algorithms. Lattice QCD required more than 40 years for the hardware, algorithms, software and analytic frameworks to move from a 10^4 hypercube of SU(2) gluons [18] to present day high-precision predictions.

We have added the following text in the Section VI: *The work that we have presented here establishes a complete framework for subsequent simulations of this process in 1+1D, and forms a foundation for those in 2+1D and 3+1D. Thus, future challenges facing 1+1D simulations are now more heavily dominated by hardware engineering and circuit implementation.*

For example, Fig. 4 involves a more complex circuit on a different processor, but the experimental errors are significantly larger and it is not obvious what additional insight is gained compared to Fig. 3. Without this kind of discussion, the manuscript remains a technical demonstration rather than a scientific study.

In that spirit, it is also hard to judge which of the co-designed techniques are “fine-tuned” to the hardware, and how much of that fine-tuning can be potentially maintained at larger scale or models. This includes for example the VQE stage for state preparation and application of post selection.

(xxvi) We have interchanged Sec. V.B and Sec. V.C to align more with the flow of the co-design. We discuss how the results obtained from executing more complex circuits on Forte (Sec. VB) informed the second set of simulations performed on Forte Enterprise (Sec. VC). The larger simulations performed on Forte involved implementing time evolution with the full weak operator (including both sea- and valence-weak operators) and without eliminating small-angle unitaries. They provided results about the overall simulation that motivated a truncation to the valence weak operator structure and informed error-mitigation strategies that were used in the truncated simulation (results of which are presented in Fig. 3, now 5), for which we obtained high precision results. The text in Sec. VB has been rewritten to clarify how the initial experiments performed on Forte were a critical aspect of the codesign process that informed the second set of experiments we ran on Forte-Enterprise.

4.The presentation also needs significant improvement. For example, section 2 discusses post-selection before the model is even defined. Terms like “twirled variants”, “JW Z string” are used without explanation.

(xxvii) The referee has a valid point. To address these points we have:

1. Moved (and modified) the Hamiltonian-specific text from Sec. IIC to Sec. IIIB, leaving the overall simulation-independent text in place.
2. Modified the caption of Fig. 1 to clarify choice of ordering of fermion sites (removing the need to use “JW Z string”): *The order of the lepton and quark spatial sites has been chosen to reduce the length of Pauli strings when mapping the fermions to qubits.* We have also added clarifying sentences to Sec. IV: *An obstacle in the way of highly parallelizable circuits is the JW transformation, which adds a string of Pauli- \hat{Z} operators to all non-mass terms in the Hamiltonian. This can be overcome by first designing the circuits without the JW \hat{Z} strings, and then including them by conjugating the \hat{Z} -string-free circuits by a sequence of CZs, taking advantage of the identity shown in Fig. 2b)*
3. Added a reference to twirling in Sec. IIC and expanded on how we generate circuit variants there. Sec. V-A has an enhanced discussion of our generation of twirled variants.

The overall procedure, including state preparation, Trotterized evolution, and (mid-circuit/final) measurement can be shown schematically.

(xxviii) We agree and have added Fig. 3 showing the structure of the circuits used in our quantum simulations (state prep, time evolution and flag-based error detection). For the state preparation we would refer the referee to Fig. 5 (now 6) which shows how we prepare the lepton registers and Fig. 6 (now 7) which shows the state preparation circuit for the quark registers. The time evolution circuit segments are shown in Fig. 7 (now 8) and Fig. 8 (now 9). The error-detecting flag circuit segments are shown in Fig. 12 (now 13), Fig. 13 (now 14), and Fig. 14 (now 15).

Reorganization into a more conventional structure with Introduction, Results, and Discussion would help improve clarity and accessibility.

(xxix) Nature’s communications has recently changed their policy regarding the formatting of initial submissions: “Within reason, style and length will not directly influence consideration of a manuscript. We also do not require a

particular structure or format at first submission. If and when revisions are required, the editor will provide detailed formatting instructions at that time.”

Our manuscript has been organized in a way consistent with journal requirements.

The authors should also improve the discussion of Fig.1, both in the text and caption, clearly distinguishing differences between L , l , and physical qubits. Currently they repeat the same spatial site indices (l), which can be confusing.

(xxx) We hope to have improved the clarity of Fig. 1 by removing the labeling of the spatial sites (since they are no longer used in the main text of the current version of the manuscript), and including the staggered site index. Previously, the l indices in Fig. 1 were repeated to indicate which quarks and leptons belonged to the same spatial site.

5.The authors rely heavily on post-processing and noise simulations due to low survival rate of the experimental shots. They use it for circuit design where they assign qubits to de-bias qubit associated error dependencies ,and post select bit strings to conserve color and total charge. However, a more intuitive and transparent discussion of how these techniques influence the physical observables and overall interpretation of results is missing.

(xxxii) There appears to be some confusion surrounding this point. We do not use noisy quantum circuit simulations as part of the error mitigation, if that is what the reviewer is referring to. When compiling the collection of circuit variants, we make use of compilation patterns chosen using our *a priori* understanding of the IonQ device characteristics. This prior information includes our understanding of, for example, the expected effects of gate error rates on different mappings between circuit qubits and physical ions, the typical types of coherent errors, and the practical effects of leakage errors.

We do mention using a Monte Carlo examination of the impact of allocating resources to either using more circuit variants or more measurements per variant. This examination is limited to exploring the statistical implications of those choices with a given post-processing strategy. The Monte Carlo draws used are from simple Dirichlet prior distributions, as described in Appendix G.

The influence of all the error mitigation techniques on the physical observables is shown in Table VII. There it is seen that applying post-selection based on the total charge (PS) has the biggest effect on moving the mean values toward the noiseless values as many types of errors do not preserve this symmetry. We have added further explanatory text in the Results section discussing these points.

Furthermore, in the discussion of the result the authors claim that “debiasing with nonlinear filtering is a superior error mitigation strategy” is made without sufficient justification or comparative analysis.

(xxxiii) We have replaced this statement with one describing more precisely our observation that debiasing with nonlinear filtering performed better for this application than ODR, given the hardware limitations.

In the discussion of Fig. 3, the authors provide little insight into the choice of the Majorana mass m_M values. In the same figure caption, the authors should clearly indicate the relevant parameters of the experimental data (i.e. trotter steps) and the simulations (approximations).

(xxxiiii) We agree with the referee on this point, and have modified the text to include the Hamiltonian parameters in the main text, see (v), as well as modified figure captions to improve clarity.

In its current form, the manuscript is not suitable for Nature Communications. That said, we do believe the work has strong foundations and could become more suitable for this venue if the authors revise it substantially. A more focused and critically reflective manuscript: one that better clarifies the scope and limitations of the theoretical and experimental approach, avoids overstated claims, but highlights the actual contributions and offers insight into what future experiments or improvements are needed. This could provide real value to the quantum simulation and HEP communities.

As it stands, the work may be more appropriate for a focused journal such as Communications Physics and can be recommended for publication there with minimal revisions.

We believe that we have addressed the concerns outlined in the three referee reports, and resubmit our manuscript for publication.

Regards,

Ivan A. Chernyshev, Roland C. Farrell, Marc Illa, Martin J. Savage, Andrii Maksymov, Felix Tripier, Miguel Angel Lopez-Ruiz, Andrew Arrasmith, Yvette de Sereville, Aharon Brodutch, Claudio Girotto, Ananth Kaushik and Martin Roetteler.

-
- [1] R. C. Farrell, I. A. Chernyshev, S. J. M. Powell, N. A. Zemlevskiy, M. Illa, and M. J. Savage, Preparations for quantum simulations of quantum chromodynamics in 1+1 dimensions. II. Single-baryon β -decay in real time, *Phys. Rev. D* **107**, 054513 (2023), arXiv:2209.10781 [quant-ph].
- [2] M. González-Alonso, O. Naviliat-Cuncic, and N. Severijns, New physics searches in nuclear and neutron β decay, *Prog. Part. Nucl. Phys.* **104**, 165 (2019), arXiv:1803.08732 [hep-ph].
- [3] V. Cirigliano, A. Garcia, D. Gazit, O. Naviliat-Cuncic, G. Savard, and A. Young, Precision Beta Decay as a Probe of New Physics (2019), arXiv:1907.02164 [nucl-ex].
- [4] S. Navas *et al.* (Particle Data Group), Review of particle physics, *Phys. Rev. D* **110**, 030001 (2024).
- [5] N. Taniuchi *et al.* (The Super-Kamiokande Collaboration), Search for proton decay via $p \rightarrow e^+\eta$ and $p \rightarrow \mu^+\eta$ with a 0.37 mton-year exposure of super-kamiokande, *Phys. Rev. D* **110**, 112011 (2024).
- [6] A. Burgman (HIBEAM, NNBAR), The HIBEAM Experiment, *Particles* **8**, 6 (2025), arXiv:2412.15933 [hep-ex].
- [7] B. Pritychenko and V. Tretyak, Comprehensive review of 2β decay half-lives, *Atom. Data Nucl. Data Tabl.* **161**, 101694 (2025).
- [8] Z. Davoudi, W. Detmold, Z. Fu, A. V. Grebe, W. Jay, D. Murphy, P. Oare, P. E. Shanahan, and M. L. Wagman (NPLQCD), Long-distance nuclear matrix elements for neutrinoless double-beta decay from lattice QCD, *Phys. Rev. D* **109**, 114514 (2024), arXiv:2402.09362 [hep-lat].
- [9] M. Agostini, G. Benato, J. A. Detwiler, J. Menéndez, and F. Vissani, Toward the discovery of matter creation with neutrinoless $\beta\beta$ decay, *Rev. Mod. Phys.* **95**, 025002 (2023), arXiv:2202.01787 [hep-ex].
- [10] V.-A. Sevestrean and S. Stoica, Theoretical Advances in Beta and Double-Beta Decay, *Symmetry* **16**, 390 (2024).
- [11] M. Horoi, Improved Statistical Analysis for the Neutrinoless Double-Beta Decay Matrix Element of ^{136}Xe , *Universe* **10**, 252 (2024).
- [12] J. M. Yao, B. Bally, J. Engel, R. Wirth, T. R. Rodríguez, and H. Hergert, Ab initio treatment of collective correlations and the neutrinoless double beta decay of ^{48}Ca , *Phys. Rev. Lett.* **124**, 232501 (2020).
- [13] D. Castillo, L. Jokiniemi, P. Soriano, and J. Menéndez, Neutrinoless $\beta\beta$ decay nuclear matrix elements complete up to N2LO in heavy nuclei, *Phys. Lett. B* **860**, 139181 (2025), arXiv:2408.03373 [nucl-th].
- [14] A. Belley, J. M. Yao, B. Bally, J. Pitcher, J. Engel, H. Hergert, J. D. Holt, T. Miyagi, T. R. Rodríguez, A. M. Romero, S. R. Stroberg, and X. Zhang, Ab initio uncertainty quantification of neutrinoless double-beta decay in ^{76}Ge , *Phys. Rev. Lett.* **132**, 182502 (2024).
- [15] V. Cirigliano, W. Dekens, and S. Urrutia Quiroga, Neutrino-less double beta decay in the ν Standard Model (2025), arXiv:2505.09679 [hep-ph].
- [16] R. C. Farrell, M. Illa, A. N. Ciavarella, and M. J. Savage, Quantum simulations of hadron dynamics in the Schwinger model using 112 qubits, *Phys. Rev. D* **109**, 114510 (2024), arXiv:2401.08044 [quant-ph].
- [17] M. Gockeler, R. Horsley, B. Klaus, D. Pleiter, P. E. L. Rakow, S. Schaefer, A. Schafer, and G. Schierholz, A lattice evaluation of four - quark operators in the nucleon, *Nucl. Phys. B* **623**, 287 (2002), arXiv:hep-lat/0103038.
- [18] M. Creutz, Monte carlo study of quantized su(2) gauge theory, *Phys. Rev. D* **21**, 2308 (1980).

Below we additionally address the remaining comments from the referees.

Referee 1:

However, I want to point out that the added paragraph in the discussion section about the IonQ trapped-ion roadmap seems a bit out of place, because it is not connected to the rest of the paper. For example, if the added paragraph talks about 800 logical qubits with $1e-7$ error rate in 2027, I would also suggest to add what kind neutrinoless double beta decay lattice model could be achieved with it.

We have re-organized the discussion section to better integrate the IonQ roadmap into the surrounding paragraphs. In addition, we have included more detail about how expected improvements in quantum computer capabilities will enable more advanced simulations of neutrinoless double beta decay.

Referee 2:

The three referees concurred to ask for details regarding approximations and fine-tunings and to make sure that existing limitations are stated more clearly earlier in the main text. The authors have addressed their concerns carefully in the reply and in the text, in a way that I consider satisfactory. The significance of the work as well as its limitations is clarified. The new figures complement the revised text efficiently.

An impressive set of state-of-the-art techniques for quantum computing has been used. It is well documented and will be a standard reference to assess the progress in this direction in the coming years. The approximations and fine-tunings put the possible observation of lepton number violation in perspective. The possible existence of a Majorana mass term is a central question in neutrino physics.

The current manuscript meets the criteria of Nature Communications. It represents a first step towards a quantum computation of the neutrinoless beta decay in actual nuclei and it will be a standard reference in the coming years.

We thank the referee for the positive comments and are pleased that the revised version satisfactorily addresses their previous concerns.

Referee 3:

These approximations are then used to underpin claims about realization of meaningful results in 2+1D and 3+1D, along with scalability to larger systems- that, in our view, cannot be justified without the systematic analysis that is missing in the paper. Moreover, the manuscript's own results suggest that the Trotterized evolution is only reliable at very short times (unless many steps are applied), which further limits the scope of the conclusions.

The referee is correct to point out that we have not provided a completely general simulation framework that will immediately allow for weak decays to be simulated in 2+1D and 3+1D once quantum computers become sufficiently capable. The simulation approach we have developed and employed serves to pioneer a new application of quantum computers to high-energy and nuclear physics that may one day impact experimental searches for new physics. In particular, our approach of performing real-time simulations of exotic weak decays directly from lattice QCD using effective weak operators and a Majorana mass terms is a general strategy that will continue to be used in future, more sophisticated quantum simulations.

Regarding scalability, an important challenge that still needs to be solved in more generality is the preparation of the initial nucleus. This poses a significant challenge due the hierarchy of energy scales relevant to realistic nuclei. This separation of scales poses problems even in classical lattice QCD simulations and is expected to also be difficult

for quantum computers. To address this we have added the following sentences to paragraph 6 of the discussion:

“Similarly, another challenge will be the robust preparation of the initial-state nucleus in a large spatial volume. This is a difficult problem, even in classical lattice QCD simulations [97–99], due to the excitation energies of nuclei being orders of magnitude smaller than their mass.”

Regarding the Trotterized time evolution, in this work we kept the number of steps fixed (varying the step size), which naturally leads to better agreement for earlier times. We do not consider this a limitation of the method, just a limitation of the hardware.

Best regards,

Ivan A. Chernyshev, Roland C. Farrell, Marc Illa, Martin J. Savage, Andrii Maksymov, Felix Tripier, Miguel Angel Lopez-Ruiz, Andrew Arrasmith, Yvette de Sereville, Aharon Brodutch, Claudio Girotto, Ananth Kaushik and Martin Roetteler.